# I2Mole: Interaction-aware Invariant Molecular Relational Learning For Generalizable Drug-Drug Interaction Prediction

**Wenjie Du**[1,2]    **Jiahui Zhang**[1,2]    **Xuqiang Li**[1,2]    **Sihan Wang**[1,2]    **Hongxin Xiang**[4]    **Jun Xia**[5]
**Ye Wei**[6]    **Yang Wang**[1,2,3*]

[1] School of School of Software Engineering, University of Science and Technology of China (USTC)
[2] Suzhou Institute for Advanced Research, USTC
[3] State Key Laboratory of Precision and Intelligent Chemistry, USTC
[4] Hunan University    [5] Westlake University    [6] City university of Hong Kong
{duwenjie, kongping, xuqiangli}@mail.ustc.edu.cn;  yeweiastronomer@gmail.com
xianghx@hnu.edu.cn  xiajun@westlake.edu.cn  angyan@ustc.edu.cn

## Abstract

Molecular interactions are a common phenomenon in physical chemistry, often resulting in unexpected biochemical properties adverse to human health, such as drug-drug interactions. Machine learning has shown great potential for predicting these interactions rapidly and accurately. However, the complexity of molecular structures and the diversity of interactions often reduce prediction accuracy and hinder generalizability. Identifying core invariant substructures (i.e., rationales) has become essential to improving the model's interpretability and generalization. Despite significant progress, existing models frequently overlook the pairwise molecular interactions, leading to insufficient capture of interaction dynamics. To address these limitations, we propose I2Mole (Interaction-aware Invariant Molecular learning), a novel framework for generalizable drug-drug interaction prediction. I2Mole meticulously models atomic interactions by first establishing indiscriminate connections between intermolecular atoms, which are then refined using an improved graph information bottleneck theory tailored for merged graphs. To further enhance model generalization, we construct an environment codebook by environment subgraph of the merged graph. This approach not only could provide noise source for optimizing mutual information but also preserve the integrity of chemical semantic information. By comprehensively leveraging the information inherent in the merged graph, our model accurately captures core substructures and significantly enhances generalization capabilities. Extensive experimental validation demonstrates I2Mole's efficacy and generalizability. The implementation code is available at https://anonymous.4open/r/I2Mol-C616.

## 1 Introduction

The molecular interaction process can give rise to additional physical or chemical properties when two or more molecules are combined (Varghese & Mushrif, 2019; Chen, 2025; Low et al., 2022a; Chen & Shi, 2025). This phenomenon is common in the fields of physics, chemistry, and medicine *etc.*, such as changes in Gibbs free energy during dissolution (*i.e.*, solute-solvent pair) (Chung et al., 2022a; Fang et al., 2024; Xia et al., 2023) and synergistic or adverse reactions between drugs (*i.e.*, drug-drug pairs) (Lee et al., 2023b; Klemperer, 1992). Due to the complexity of molecular structures and the diversity of molecular interactions, conventional modeling approaches are limited and susceptible to noise, undermining prediction accuracy. Meanwhile, they lack generalizability and reliability severely limits their applicability. Based on this, mining the invariant core substructures of molecules (*i.e.*, rationale) has become a widely accepted strategy to enhance both interpretability and generalization like the CGIB (Lee et al., 2023a) and MoleOOD (Yang et al., 2022b).

---

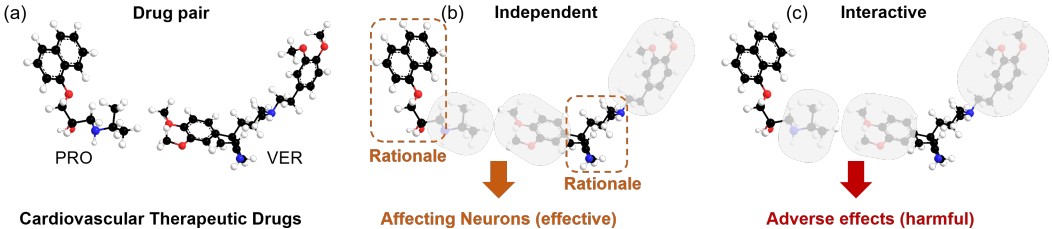

Figure 1: An Example of molecular interactions. (a) Propranolol (PRO) and Verapamil (VER) are widely prescribed cardiovascular therapeutic agents; (b) each drug is influenced by distinct core substructures to achieve affect; (c) harmful effects on human health occur by co-administered.

Although current methods have attracted widespread attention in predicting the properties of molecular pairs, two inherent shortcomings remain underexplored. The first is **Insufficiency in molecular interaction modeling.** Existing methods demonstrate proficiency in elucidating essential structural characteristics for individual molecular models. However, when drug-drug interactions (DDI) occur, pivotal substructures may exhibit considerable variation. For example, Propranolol and Verapamil (Figure 1 (a)) are commonly prescribed drugs for the treatment of hypertension and cardiac arrhythmias, yet they act through distinct pharmacological mechanisms that affect cardiovascular function. The aromatic ring and hydroxyl groups in Propranolol are critical for receptor binding and $\beta$-adrenergic inhibition, while the phenyl rings and amino moieties in Verapamil mediate L-type calcium channel inhibition, as illustrated in Figure 1 (b). However, when co-administered, the interaction between Propranolol's $\beta$-blocking pharmacophore and Verapamil's calcium-channel–blocking substructures may excessively suppress cardiac conduction, leading to severe adverse effects such as excessive bradycardia or atrioventricular block (Figure 1 (c)). Therefore, comprehensive modeling of intermolecular interactions is crucial for a profound understanding of molecular interactions.

Some current models have noticed the aforementioned shortcomings (Behler, 2015; 2016). However, they still **lack consideration of model generalization.** Given the diverse and complex nature of molecular species in real-world scenarios, the data used for training and testing may inevitably be sampled from different distributions, thus presenting challenges related to OOD (Paul et al., 2021; Petrova, 2013; Yang et al., 2022b). While introducing integrated noise injection techniques to simulate diverse environmental distributions holds promise for enhancing model generalization and capturing core rationales, several drawbacks exist. Specifically, 1) The simulation of noise data may fail to accurately reflect authentic environmental vectors in chemical space. 2) Indiscriminate noise injection can distort semantic information and hinder model convergence, while random environmental vectors may inadequately represent the broad distribution of molecular interactions; and 3) when the injected noise variance is too small, the noise effect may vanish, defeating its intended purpose.

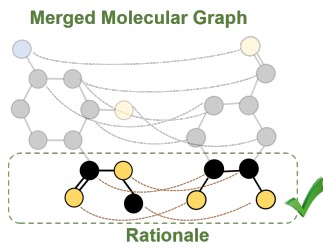

Figure 2: Diagram illustrating molecular interaction modeling to capture rationales. Molecular pairs will be constructed into a merged graph by connecting atoms pairwise (dashed lines). Please note that to avoid excessive complexity, some unimportant relation edges will be removed (unconnected)

In light of this, we introduce an **I**nteraction-aware **I**nvariant **Mole**cular learning framework, termed I2Mole, for generalizable DDI prediction. Spontaneous molecular interaction phenomena, tend to occur in specific molecular structures (*e.g.*, -OH, =O, N), giving rise to stronger intermolecular interactions. We carefully design dynamic weighted relational edges to model the atom–atom interaction relationships. Conversely, for atomic pairwise interactions that rarely occur, we employ iterative truncation to restrict their message passing processes, thereby reducing interference with the overall learning of the merged graph while moderately lowering graph complexity, as presented in Figure 2. Given the vastness and largely unexplored nature of the chemical space, we further introduce the concept of vector quantization (VQ) (van den Oord et al., 2017; Razavi et al., 2019) for molecular interactions to construct a merged graph environment codebook. This codebook clusters the potential environments of

molecules in the training set into a predefined number of categories, and the learned environmental distribution also serves as a controllable noise source for mutual information optimization (Duncan, 1970; Yu et al., 2022b). Therefore, our I2Mole which incorporates explicit molecule interactions and an improved environment codebook, effectively achieves generalizable property prediction on various DDI datasets.

## 2 PRELIMINARIES

### 2.1 PROBLEM FORMULATION

A molecule can be depicted as a graph $\mathcal{G}$ whose nodes $\mathcal{V}$ denote the atoms and edges $\mathcal{E}$ act as the bonds Wen et al. (2021). $\mathcal{U}$ is the global feature vector which is extracted from each molecule (Appendix D). Given a set of drug molecular graph pairs $\mathcal{D} = \left\{ (\mathcal{G}_a^1, \mathcal{G}_b^1), (\mathcal{G}_a^2, \mathcal{G}_b^2), \ldots, (\mathcal{G}_a^n, \mathcal{G}_b^n) \right\}$ and their associated target values $\mathbb{Y} = \left\{ \mathbf{Y}^1, \mathbf{Y}^2, \ldots, \mathbf{Y}^n \right\}$, our objective is to train a model $\mathcal{M}$ that can classify the target values for arbitrary drug pairs in an end-to-end manner, *i.e.*, $\mathbf{Y}^i = \mathcal{M}(\mathcal{G}_a^i, \mathcal{G}_b^i)$.

### 2.2 GRAPH INFORMATION BOTTLENECK (GIB)

According to the GIB principle (Yu et al., 2020; 2022b; Miao et al., 2022), we could get:

$$\mathcal{G}_{\text{IB}} = \underset{\mathcal{G}_{\text{sub}} \in \mathcal{S}}{\arg\min} - I(\mathbf{Y}; \mathcal{G}_{\text{sub}}) + \beta I(\mathcal{G}; \mathcal{G}_{\text{sub}}). \quad (1)$$

Intuitively, $\mathcal{S}$ represents the set of $\mathcal{G}_{\text{sub}}$, and $\mathcal{G}_{\text{IB}}$ is the core subgraph of $\mathcal{G}$, which discards information by minimizing the mutual information $I(\mathcal{G}; \mathcal{G}_{\text{sub}})$, while preserving target-relevant information by maximizing the mutual information $I(\mathbf{Y}; \mathcal{G}_{\text{sub}})$.

### 2.3 INVARIANT LEARNING

Given the distribution shift between training and testing data, recent studies (Rojas-Carulla et al., 2018a; Arjovsky et al., 2019; Wu et al., 2022a) propose the existence of a potential environment variable **env** to express this problem:

$$\min_{f} \max_{\mathcal{G}_{\text{env}} \in \mathbf{E}} \mathbb{E}_{(\mathcal{G}, Y) \sim p(\mathcal{G}, \mathbf{Y} | \mathbf{env} = \mathcal{G}_{\text{env}})} [R(f(\mathcal{G}), \mathbf{Y}) \mid \mathcal{G}_{\text{env}}], \quad (2)$$

where $\mathbf{E}$ denotes the environment support, $f(\cdot)$ represents the predictive model, and $R(\cdot, \cdot)$ is the risk function. The label $\mathbf{Y}$ is independent of the environment $\mathcal{G}_{\text{env}}$, conditioned on the subgraph $\mathcal{G}_{\text{sub}}$:

$$\mathbf{Y} \perp \mathcal{G}_{\text{env}} \mid \mathcal{G}_{\text{sub}}, \quad (3)$$

where $\perp$ denotes probabilistic independence. These principles collectively protect predictions from external influences, ensuring that the rationale comprehensively captures all discriminative features. This is for a single molecule, and we would extend it to molecular pairs.

## 3 METHODOLOGY

In this section, we detail our proposed method. In Section 3.1, we define the merged graph and the intermolecular message passing mechanism. Section 3.2 explains the details of subgraph extraction by GIB theory. In Section 3.3, we describe how to inject environmental embeddings into the rationales to enhance model generalization. Section 3.4 presents the total loss function of I2Mole.

### 3.1 MERGED MOLECULAR REPRESENTATION

**Molecule Merging.** The merged graph $\widetilde{\mathcal{G}}$ could be generated by establishing a weighted relational edge between two molecules which connects each atom pairwise.

$$\widetilde{\mathcal{G}} = \{\mathcal{R}, \mathcal{E}, \mathcal{V}, \mathcal{U}\}. \quad (4)$$

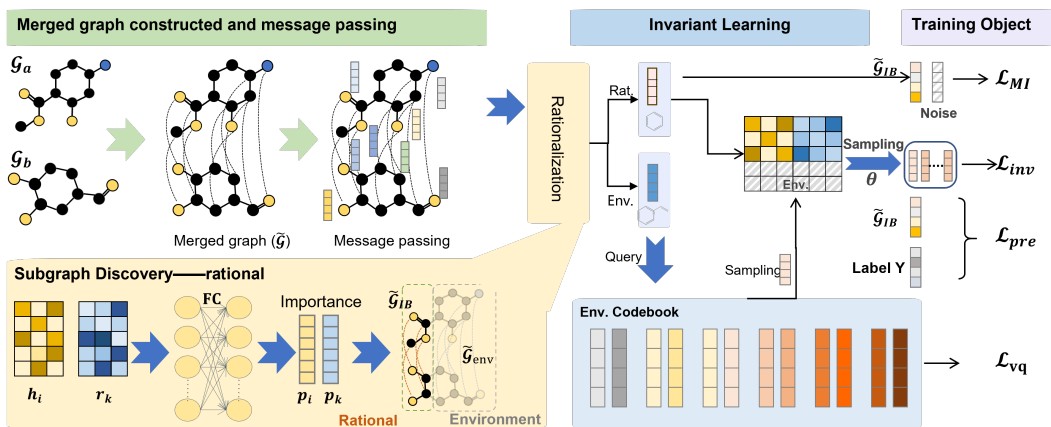

Figure 3: Overview of our model. Initially, molecular pairs construct a merged graph to facilitate the message passing process. Subsequently, subgraphs are extracted based on the GIB, and the environmental components are recorded in a codebook. During the invariant learning process, the rationale part concatenates different environment embeddings to achieve invariant representations.

The set of relation edges are $\mathcal{R}$:

$$\mathcal{R} = \{(r_{ij}, v_{ai}, v_{bj})\}_{k=1}^{N^a \times N^b}, \tag{5}$$

where $ai \in \{1, 2, 3, \ldots, N^a\}$, $bj \in \{1, 2, 3, \ldots, N^b\}$. $N^a$ and $N^b$ are the total number of atoms in drug molecular graph $\mathcal{G}_a$, $\mathcal{G}_b$. $r_{ij}$ represents the relation edge. And $k$ is the index of $\mathcal{R}$.

**Intra-molecular message passing.** Generally, in this merged molecular graph $\widetilde{\mathcal{G}}$, the message passing process is first executed intramolecule. In this process, $e_{ij}$ is updated to $e'_{ij}$ by aggregating the initial bond features, and the two atomic features, $v_i$ and $v_j$, and the global features $u$. In addition, the feature vector $v_i$ and $u$ are updated to $v'_i$ and $u'_i$ respectively:

$$e'_{ij} = e_{ij} + \text{LeakyReLU}[\text{FC}(v_i + v_j) + [\text{FC}(e_{ij}) + [\text{FC}(u)]], \tag{6}$$

$$\hat{e}_{ij} = \frac{\sigma(e'_{ij})}{\sum_{j' \in N_i} \sigma(e'_{ij'}) + \epsilon}, \quad v'_i = v_i + \text{LeakyReLU}[\text{FC}(v_i + \sum_{j \in N_i} \hat{e}_{ij} \odot \text{FC}(v_j)) + \text{FC}(u)], \tag{7}$$

$$u' = u + \text{LeakyReLU}[\text{FC}(\frac{1}{N^v} \sum_{i=1}^{N^v} v'_i + \frac{1}{N^e} \sum_{k=1}^{N^e} e'_k + u)], \tag{8}$$

where FC is a fully connected layer. $\odot$ denotes the Hadamard product. $\sigma(\cdot)$ is sigmoid activation function, and $\epsilon$ is a fixed constant (0.0001). $N^v$ and $N^e$ are the number of atoms and bonds.

**Intermolecular message passing.** We utilize GAT network(Veličković et al., 2017) for intermolecular message passing to calculate the weight of the relation edge $r_{ij}$.

$$r_{ij} = \text{LeakyReLU}(\text{FC}(\mathbf{W}v'_{ai}, \mathbf{W}v'_{bj})), \tag{9}$$

where $W$ is the learnable weight matrix. Based on the calculated attention coefficients ($r_{ij}$) for the relation edge, we perform global sorting and retain the top_x% (a hyperparameter):

$$r'_{ij} = \begin{cases} r_{ij} & \text{if } r_{ij} \geq X, \\ 0 & \text{otherwise.} \end{cases} \tag{10}$$

Here, $X$ represents the threshold corresponding to the top_x% ranking of $r_{ij}$ values. The selected attention coefficients are then normalized across the entire graph to facilitate the intermolecular information-passing process. The atomic feature $v'_{ai}$ for $i$ is updated, also for $v'_{bj}$:

$$\alpha_{ij} = \frac{r'_{ij}}{\sum_{i,j} r'_{ij}}, \quad v''_{ai} = (1 - \sum_{j \in N_b} \alpha_{ij})v'_{ai} + \sum_{j \in N_b} \alpha_{ij}v'_{bj}, \tag{11}$$

## 3.2 CORE SUBSTRUCTURE EXTRACTION BASED ON GIB

We optimize the objective Equation 12 to detect the core structure in the merged graph:

$$\widetilde{\mathcal{G}}_{\text{IB}} = \underset{\widetilde{\mathcal{G}}_{\text{sub}} \in \widetilde{\mathcal{S}}}{\arg\min} - I(\mathbf{Y}; \widetilde{\mathcal{G}}_{\text{sub}}) + \beta I(\mathcal{G}; \widetilde{\mathcal{G}}_{\text{sub}}), \tag{12}$$

where $\widetilde{\mathcal{S}}$ represents the set of $\widetilde{\mathcal{G}}_{\text{sub}}$. Each term indicates the prediction and compression terms respectively, which should be minimized during training, as outlined below.

### 3.2.1 EXTRACT TARGET-ORIENTED INFORMATION

Minimizing $-I(\mathbf{Y}; \widetilde{\mathcal{G}}_{\text{IB}})$, which is to calculate upper bound of $-I(\mathbf{Y}; \widetilde{\mathcal{G}}_{\text{IB}})$. Given the merged graph $\widetilde{\mathcal{G}}$, its label information $\mathbf{Y}$, and the learned IB-graph $\widetilde{\mathcal{G}}_{IB}$, we have:

$$-I(\mathbf{Y}; \widetilde{\mathcal{G}}_{\text{IB}}) \le \mathbb{E}_{\mathbf{Y}; \widetilde{\mathcal{G}}_{\text{IB}}}[-\log p_\theta(\mathbf{Y}|\widetilde{\mathcal{G}}_{\text{IB}})] := \mathcal{L}_{pre}, \tag{13}$$

where $p_\theta(\mathbf{Y}|\widetilde{\mathcal{G}}_{\text{IB}})$ is variational approximation of $p(\mathbf{Y}|\widetilde{\mathcal{G}}_{\text{IB}})$. $p_\theta(\mathbf{Y}|\widetilde{\mathcal{G}}_{\text{IB}})$ is a predictor parametrized by $\theta$. Thus, we can minimize the upper bound of $-I(\mathbf{Y}; \widetilde{\mathcal{G}}_{\text{IB}})$ by minimizing the model prediction loss $\mathcal{L}_{\text{pre}}(\mathbf{Y}, \widetilde{\mathcal{G}}_{\text{IB}})$ with cross-entropy loss. Proofs are in Appendix C.1 (Sufficiency assumption (Yang et al., 2022b)).

### 3.2.2 OPTIMIZE MINIMIZED $\widetilde{G}$

Minimizing $I(\widetilde{\mathcal{G}}; \widetilde{\mathcal{G}}_{\text{IB}})$, which is to calculate upper bound of $I(\widetilde{\mathcal{G}}; \widetilde{\mathcal{G}}_{\text{IB}})$. Inspired by a recent approach on graph information bottleneck (Yu et al., 2022b), we also minimize $I(\widetilde{\mathcal{G}}; \widetilde{\mathcal{G}}_{\text{IB}})$ by injecting noise into node representations. Then, we dampen the information in $\widetilde{\mathcal{G}}$ by injecting noise into node representations with a learned probability. Let $\epsilon$ be the noise sampled from a parametric noise distribution. We assign each node a probability of being replaced by $\epsilon$. Specifically, for the $i$-th node, the $k$-th relation edge, we learn the probability $p_i$ and $p_k$ using a fully connected layer. Then, we apply a Sigmoid function on the output of fully connected layer to ensure $p_i, p_k \in [0, 1]$:

$$p_i = \text{Sigmoid}(\text{FC}(h_i)), \quad p_k = \text{Sigmoid}(\text{FC}(\text{r}_k)). \tag{14}$$

Next, if a $k$-th relation edge is connected to the $i$-th node, we adjust the probability $p_i$ by adding $\frac{p_k}{N}$ to it, where $N$ depends on whether the $i$-th node is in $\mathcal{G}a$ or $\mathcal{G}b$:

$$p_i = \begin{cases} p_i + \frac{p_k}{N_b} & \text{if } i \in \mathcal{G}_a \text{ and } k\text{-th edge is connected to } i\text{-th node}, \\ p_i + \frac{p_k}{N_a} & \text{if } i \in \mathcal{G}_b \text{ and } k\text{-th edge is connected to } i\text{-th node}. \end{cases} \tag{15}$$

We then replace the node representation $h_i$ by $\epsilon$ with probability $p_i$:

$$z_i = \lambda_i h_i + (1 - \lambda_i)\epsilon, \quad \mathbf{h}_i^r = (1 - \lambda_i)\mathbf{h}_i, \tag{16}$$

where $\lambda_i \sim \text{Bernoulli}(p_i)$, $\mathbf{h}_i^r$ is the irrelevant substructure node which would be used to construct $\widetilde{\mathcal{G}}_{\text{env}}$. The transmission probability $p_i$ controls the information sent from $h_i$ to $z_i$. If $p_i = 1$, then all the information in $h_i$ is transferred to $z_i$ without loss. On the contrary, when $p_i = 0$, then $z_i$ contains no information from $h_i$ but only noise. We hope $p_i$ is learnable so that we can selectively preserve the information in $\widetilde{\mathcal{G}}_{\text{IB}}$. However, $\lambda_i$ is a discrete random variable and we cannot directly calculate the gradient of $p_i$. Therefore, we employ the concrete relaxation (Jang et al., 2016) for $\lambda_i$:

$$\lambda_i = \text{Sigmoid}(\frac{1}{t} \log \frac{p_i}{1 - p_i} + \log \frac{u}{1 - u}), \tag{17}$$

where $t$ is the temperature parameter and $u \sim \text{Uniform}(0, 1)$. Another critical aspect of noise injection is the characterization of the injected noise. It is important that arbitrary noise can be detrimental to the semantic integrity of the input graph, leading to predictions that deviate from the

actual graph properties. Conversely, appropriately selected noise can provide a variational upper bound to the overall objective. Therefore, the minimizing the upper bound of $I(\widetilde{\mathcal{G}}_{\text{IB}}; \widetilde{\mathcal{G}})$ as follows:

$$I(\widetilde{\mathcal{G}}_{\text{IB}}; \widetilde{\mathcal{G}}) \leq \mathbb{E}_{\mathcal{G}} \left[ -\frac{1}{2} \log A_{\widetilde{\mathcal{G}}} + \frac{1}{2m_{\widetilde{\mathcal{G}}}} A_{\widetilde{\mathcal{G}}} + \frac{1}{2m_{\widetilde{\mathcal{G}}}} B_{\widetilde{\mathcal{G}}}^2 \right] := \mathcal{L}_{\text{MI}}(\widetilde{\mathcal{G}}_{\text{IB}}, \widetilde{\mathcal{G}}), \tag{18}$$

where $A_{\widetilde{\mathcal{G}}} = \sum_{j=1}^{m_{\widetilde{\mathcal{G}}}} (1 - \lambda_j)^2$ and $B_{\widetilde{G}} = \frac{\sum_{j=1}^{m_{\widetilde{\mathcal{G}}}} \lambda_j (h_j - \mu_h)}{\sigma_h}$. More details are given in Appendix C.2.

### 3.3 ENVIRONMENT INFERENCE

Based on the above steps, we can identify the decisive core substructure $\widetilde{\mathcal{G}}_{\text{IB}}$ in Equation 12. However, relying solely on $\widetilde{\mathcal{G}}_{\text{IB}}$ may not ensure robust generalization across diverse distributions. To enhance its robustness, we incorporate principles from invariance learning theory, integrating features from various environments encountered across diverse distributions. The problem definition is as follows:

$$\min_f \max_{\widetilde{\mathcal{G}}_{\text{env}} \in \mathbf{E}} \mathbb{E}_{(\widetilde{\mathcal{G}}, \mathbf{Y}) \sim q(\widetilde{\mathcal{G}}_{\text{env}})} [R(f(\widetilde{\mathcal{G}}), \mathbf{Y}) \mid \widetilde{\mathcal{G}}_{\text{env}}], \tag{19}$$

where $\mathbf{E}$ denotes the support of environments. The irrelevant substructures $\widetilde{\mathcal{G}}_{\text{env}}$ can be viewed as the environment, with each node embedding being $\mathbf{h}_i^r$. $q(\widetilde{\mathcal{G}}_{\text{env}})$ is the distribution of data under environment $\widetilde{\mathcal{G}}_{\text{env}}$ combined with various rationales, $f(\cdot)$ is the prediction model and $R(\cdot, \cdot)$ is the risk function such as cross-entropy loss. Equation 19 aims to minimize the maximum errors across different environments, thus guaranteeing the capture of invariance across environments (Wu et al., 2022c;b).

Directly solving Equation 19 is impractical due to limited training data across the various environments in $\mathbf{E}$. Here, we introduce VQ van den Oord et al. (2017); Razavi et al. (2019) to create a trainable environment codebook $W = \{env_1, env_2, \ldots, env_M\}$, defining a latent embedding space $env \in \mathbb{R}^{M \times F}$. Here, $M$ represents the number of discrete environments (*i.e.*, $env$), and $F$ denotes the dimension of each latent vector. A nearest neighbor lookup is used in the shared embedding space $\mathbf{E}$ to find the closest latent vector $env_m$, indexed by $m$. Additionally, the set2set network (Vinyals et al., 2015) is utilized to pool $\widetilde{\mathcal{G}}_{\text{IB}}, \widetilde{\mathcal{G}}_{\text{env}}, \widetilde{\mathcal{G}}$, resulting in the substructure representation vectors $\widetilde{s}_{\text{IB}}$, $\widetilde{s}_{\text{env}}$ and $\widetilde{s}_{\mathcal{G}}$. This process acts as a specific non-linearity that maps the latent vectors $\widetilde{s}_{\text{env}}$ to one of the $M$ embedding vectors:

$$q(m \mid \widetilde{s}_{env}) = \begin{cases} 1 & \text{for } m = \arg\min_j \|\widetilde{s}_{\text{env}} - env_j\|_2, \\ 0 & \text{otherwise.} \end{cases} \tag{20}$$

To update the codebook and encourage the output of the encoder to stay close to the chosen codebook embedding, where the sg[·] denotes the stop-gradient and $\delta$ is set to 0.25 Xia et al. (2022):

$$\mathcal{L}_{vq} = \|\text{sg}[\widetilde{s}_{\text{env}}] - env_m\|_2^2 + \delta \|\widetilde{s}_{\text{env}} - \text{sg}[env_m]\|_2^2. \tag{21}$$

As $\mathcal{L}_{vq}$ gradually converges, we obtain a stable codebook set $W$, which clusters the infinite possible environment space $\mathbf{E}$ into a discretized set of $M$ finite environments represented by $W$. Subsequently, we traverse all potential environment vectors ($env$) and assign rationales to different environments to achieve stable predictions. This ensures that the prediction results of the rationales are independent, thereby guaranteeing the independence (Invariance assumption Yang et al. (2022b)).

$$\min_f \mathbb{E}_{env_i \in W} \mathbb{E}_{(\widetilde{s}_{\mathcal{G}}, \mathbf{Y}) \sim q(env_i)} [R(f(\widetilde{s}_{\mathcal{G}}), \mathbf{Y}) \mid env_i]. \tag{22}$$

This formula can be obtained by minimizing the weighted sum of cross-entropy losses across different environments. Assuming a total of $C$ classes, let $\phi_i$ denote the probability that $env$ belongs to $env_i$, and let $\Phi$ represents the classification head that maps the molecular representation to the category labels. the encoder $f_{env}$ and the classification head $\Phi$ together form the prediction model. So, the loss can be expressed in the following form, where ‖ denotes the concatenation operation:

$$\mathcal{L}_{inv} = -\sum_{i=1}^{M} \phi_i \sum_{\widetilde{s}_{\mathcal{G}} \in \mathcal{D}_{\text{train}}} \sum_{c=1}^{C} \mathbf{Y}_{\widetilde{s}_{\mathcal{G}}} \log \Phi \left( f_{env}(\widetilde{s}_{\text{IB}} \| env_i) \right). \tag{23}$$

Table 1: Performance of different methods in transductive setting. (Bold numbers are the best results, while the top-performing baseline is superscript cross. The standard deviations is in parentheses).

| | ZhangDDI | | | ChchMiner | | | DeepDDI | | |
|---|---|---|---|---|---|---|---|---|---|
| | ACC ($\uparrow$) | AUROC ($\uparrow$) | F1 ($\uparrow$) | ACC ($\uparrow$) | AUROC ($\uparrow$) | F1 ($\uparrow$) | ACC ($\uparrow$) | AUROC ($\uparrow$) | F1 ($\uparrow$) |
| DeepDDI | $83.35_{(0.49)}$ | $91.13_{(0.58)}$ | $80.24_{(0.47)}$ | $90.34_{(0.44)}$ | $95.71_{(0.37)}$ | $91.83_{(0.28)}$ | $92.39_{(0.38)}$ | $95.10_{(0.42)}$ | $91.32_{(0.39)}$ |
| SSI-DDI | $86.97_{(0.62)}$ | $93.76_{(0.34)}$ | $82.99_{(0.30)}$ | $93.26_{(0.24)}$ | $97.81_{(0.25)}$ | $93.11_{(0.19)}$ | $94.27_{(0.25)}$ | $97.42_{(0.31)}$ | $95.41_{(0.19)}$ |
| MDF-SA-DDI | $86.89_{(0.15)}$ | $94.03_{(0.22)}$ | $83.67_{(0.14)}$ | $94.63_{(0.21)}$ | $98.10_{(0.17)}$ | $94.17_{(0.16)}$ | $94.12_{(0.21)}$ | $88.84_{(0.26)}$ | $96.13_{(0.17)}$ |
| DSN-DDI | $87.65_{(0.13)}$ | $94.63_{(0.18)}$ | $84.30_{(0.09)}$ | $94.25_{(0.11)}$ | $98.31_{(0.10)}$ | $95.34_{(0.08)}$ | $95.74_{(0.18)}$ | $98.06_{(0.16)}$ | $96.71_{(0.11)}$ |
| CGIB | $87.32_{(0.71)}$ | $94.43_{(0.60)}$ | $84.53_{(0.45)}$ | $94.37_{(0.39)}$ | $98.38_{(0.31)}\dagger$ | $95.44_{(0.24)}$ | $95.76_{(0.72)}$ | $98.08_{(0.64)}\dagger$ | $96.53_{(0.53)}$ |
| CMRL | $87.78_{(0.37)}$ | $94.08_{(0.23)}$ | $84.78_{(0.25)}$ | $94.43_{(0.25)}$ | $98.37_{(0.12)}$ | $95.62_{(0.17)}$ | $95.49_{(0.34)}$ | $98.03_{(0.31)}$ | $96.82_{(0.29)}\dagger$ |
| IE-HGNN | $86.93_{(0.18)}$ | $94.32_{(0.23)}$ | $84.93_{(0.12)}$ | $94.48_{(0.28)}$ | $98.36_{(0.19)}$ | $95.57_{(0.18)}$ | $95.57_{(0.22)}$ | $97.98_{(0.23)}$ | $96.58_{(0.20)}$ |
| IGIB-ISE | $88.08_{(0.26)}\dagger$ | $94.71_{(0.18)}\dagger$ | $85.39_{(0.17)}\dagger$ | $94.92_{(0.21)}\dagger$ | $98.24_{(0.14)}$ | $95.84_{(0.16)}\dagger$ | $95.85_{(0.19)}\dagger$ | $98.02_{(0.20)}$ | $96.71_{(0.15)}$ |
| Ours | $\mathbf{88.64}_{(0.24)}$ | $\mathbf{95.12}_{(0.12)}$ | $\mathbf{85.87}_{(0.20)}$ | $\mathbf{95.34}_{(0.19)}$ | $\mathbf{98.84}_{(0.10)}$ | $\mathbf{96.21}_{(0.25)}$ | $\mathbf{96.51}_{(0.14)}$ | $\mathbf{99.04}_{(0.22)}$ | $\mathbf{97.53}_{(0.15)}$ |

Table 2: Performance of different methods in inductive settings. (Bold numbers are the best results, while the top-performing baseline is superscript cross. The standard deviations is in parentheses).

| Type1 | | | | | | | | | |
|---|---|---|---|---|---|---|---|---|---|
| | ZhangDDI | | | ChchMiner | | | DeepDDI | | |
| | ACC ($\uparrow$) | AUROC ($\uparrow$) | F1 ($\uparrow$) | ACC ($\uparrow$) | AUROC ($\uparrow$) | F1 ($\uparrow$) | ACC ($\uparrow$) | AUROC ($\uparrow$) | F1 ($\uparrow$) |
| DeepDDI | $60.84_{(1.34)}$ | $59.51_{(1.18)}$ | $43.81_{(1.26)}$ | $66.19_{(1.08)}$ | $68.51_{(1.53)}$ | $67.67_{(1.29)}$ | $64.39_{(1.71)}$ | $69.52_{(1.53)}$ | $68.31_{(1.45)}$ |
| SSI-DDI | $62.38_{(1.53)}$ | $69.56_{(1.21)}$ | $47.59_{(1.17)}$ | $76.94_{(1.32)}$ | $79.64_{(1.53)}$ | $77.61_{(1.24)}$ | $69.77_{(0.86)}$ | $75.93_{(1.14)}$ | $72.23_{(0.77)}$ |
| MDF-SA-DDI | $64.51_{(1.39)}$ | $70.99_{(1.27)}$ | $51.53_{(1.15)}$ | $75.39_{(0.80)}$ | $80.47_{(0.68)}$ | $79.83_{(1.05)}$ | $71.13_{(0.77)}$ | $80.54_{(0.94)}$ | $71.61_{(0.88)}$ |
| DSN-DDI | $67.68_{(0.87)}$ | $72.49_{(1.02)}$ | $53.64_{(0.77)}$ | $78.94_{(0.72)}$ | $85.93_{(0.65)}$ | $83.81_{(0.83)}$ | $73.35_{(0.62)}$ | $83.11_{(0.76)}$ | $75.68_{(0.70)}$ |
| CGIB | $68.34_{(0.66)}$ | $72.80_{(0.43)}$ | $57.29_{(0.58)}\dagger$ | $79.75_{(0.73)}$ | $86.41_{(0.93)}$ | $85.13_{(0.43)}$ | $73.86_{(0.97)}$ | $80.80_{(0.53)}$ | $78.47_{(0.47)}$ |
| CMRL | $68.38_{(1.12)}$ | $74.59_{(1.05)}$ | $56.41_{(0.97)}$ | $80.54_{(0.66)}$ | $87.64_{(0.54)}$ | $86.55_{(0.57)}\dagger$ | $74.12_{(0.55)}$ | $84.96_{(0.87)}$ | $77.81_{(0.74)}$ |
| IE-HGNN | $68.24_{(0.92)}$ | $74.02_{(0.83)}$ | $56.73_{(0.88)}$ | $80.21_{(0.77)}$ | $87.92_{(0.69)}$ | $86.21_{(0.81)}$ | $73.51_{(0.64)}$ | $85.07_{(0.71)}$ | $77.42_{(0.66)}$ |
| IGIB-ISE | $68.49_{(0.87)}\dagger$ | $74.61_{(0.78)}\dagger$ | $57.10_{(0.74)}$ | $80.83_{(0.71)}\dagger$ | $88.22_{(0.64)}\dagger$ | $86.52_{(0.70)}$ | $74.32_{(0.61)}\dagger$ | $85.41_{(0.66)}\dagger$ | $78.65_{(0.58)}\dagger$ |
| Ours | $\mathbf{69.12}_{(0.23)}$ | $\mathbf{75.14}_{(0.42)}$ | $\mathbf{57.89}_{(1.55)}$ | $\mathbf{81.59}_{(1.10)}$ | $\mathbf{88.51}_{(0.31)}$ | $\mathbf{87.43}_{(0.74)}$ | $\mathbf{75.27}_{(0.64)}$ | $\mathbf{85.62}_{(0.74)}$ | $\mathbf{78.96}_{(0.37)}$ |

| Type2 | | | | | | | | | |
|---|---|---|---|---|---|---|---|---|---|
| | ZhangDDI | | | ChchMiner | | | DeepDDI | | |
| | ACC ($\uparrow$) | AUROC ($\uparrow$) | F1 ($\uparrow$) | ACC ($\uparrow$) | AUROC ($\uparrow$) | F1 ($\uparrow$) | ACC ($\uparrow$) | AUROC ($\uparrow$) | F1 ($\uparrow$) |
| DeepDDI | $58.62_{(2.03)}$ | $56.34_{(1.97)}$ | $25.19_{(4.34)}$ | $63.78_{(2.14)}$ | $66.71_{(2.67)}$ | $71.37_{(3.38)}$ | $61.68_{(4.18)}$ | $65.17_{(3.72)}$ | $66.74_{(4.16)}$ |
| SSI-DDI | $57.24_{(2.38)}$ | $59.34_{(3.26)}$ | $37.16_{(3.84)}$ | $65.61_{(2.51)}$ | $68.39_{(1.94)}$ | $74.95_{(2.17)}$ | $65.53_{(3.53)}$ | $69.37_{(4.16)}$ | $62.18_{(3.94)}$ |
| MDF-SA-DDI | $57.63_{(1.89)}$ | $55.97_{(1.67)}$ | $33.94_{(2.78)}$ | $65.24_{(1.97)}$ | $68.54_{(2.04)}$ | $77.32_{(1.89)}$ | $66.34_{(1.55)}$ | $70.81_{(2.01)}$ | $70.95_{(1.71)}$ |
| DSN-DDI | $58.37_{(1.31)}$ | $58.88_{(1.12)}$ | $39.49_{(2.32)}$ | $68.36_{(1.54)}$ | $69.34_{(1.34)}$ | $77.52_{(1.21)}$ | $68.17_{(1.28)}$ | $72.71_{(1.37)}$ | $71.96_{(1.64)}$ |
| CGIB | $58.39_{(2.04)}$ | $57.24_{(1.97)}$ | $28.83_{(4.53)}$ | $68.78_{(1.84)}$ | $69.82_{(1.39)}$ | $78.46_{(2.03)}$ | $68.26_{(1.39)}$ | $68.78_{(1.67)}$ | $75.75_{(1.75)}\dagger$ |
| CMRL | $60.78_{(1.37)}\dagger$ | $60.02_{(2.03)}\dagger$ | $38.73_{(3.04)}\dagger$ | $67.09_{(1.54)}$ | $69.62_{(1.67)}$ | $75.76_{(1.28)}$ | $68.29_{(1.78)}$ | $73.38_{(1.96)}$ | $73.91_{(2.14)}$ |
| IE-HGNN | $60.47_{(1.34)}$ | $61.18_{(1.21)}$ | $38.92_{(1.87)}$ | $68.71_{(1.06)}$ | $69.47_{(0.97)}$ | $78.92_{(1.24)}$ | $68.13_{(0.92)}$ | $73.56_{(1.02)}$ | $74.92_{(1.17)}$ |
| IGIB-ISE | $59.96_{(1.20)}$ | $59.71_{(1.12)}$ | $38.62_{(1.58)}$ | $68.92_{(0.96)}\dagger$ | $69.94_{(0.89)}\dagger$ | $79.32_{(1.05)}\dagger$ | $68.41_{(0.88)}\dagger$ | $74.10_{(0.92)}\dagger$ | $75.60_{(1.01)}$ |
| Ours | $\mathbf{61.35}_{(1.01)}$ | $\mathbf{62.02}_{(1.09)}$ | $\mathbf{39.95}_{(1.56)}$ | $\mathbf{69.23}_{(0.17)}$ | $\mathbf{70.02}_{(0.85)}$ | $\mathbf{79.67}_{(0.35)}$ | $\mathbf{69.92}_{(0.11)}$ | $\mathbf{74.27}_{(0.62)}$ | $\mathbf{75.83}_{(0.43)}$ |

## 3.4 TRAINING OBJECTIVE

Finally, we train the model with the following objective:

$$\mathcal{L}_{\text{total}} = \mathcal{L}_{\text{inv}} + \mathcal{L}_{\text{pre}} + \beta\mathcal{L}_{\text{MI}} + \gamma\mathcal{L}_{\text{vq}} \tag{24}$$

Here, $\mathcal{L}_{\text{pre}}$ and $\mathcal{L}_{\text{MI}}$ are guided by the GIB. $\mathcal{L}_{\text{pre}}$ is the cross-entropy loss for classification tasks. $\mathcal{L}_{\text{MI}}$ represents the KL divergence between the core substructures and the non-core subgraph, encouraging substructure compression. And $\mathcal{L}_{\text{inv}}$ aims to minimize the disturbance loss across various environments. $\beta$ and $\gamma$ are trade-off parameters that govern the weight of $\mathcal{L}_{\text{MI}}$ and $\mathcal{L}_{\text{vq}}$.

## 4 EXPERIMENT AND ANALYSE

### 4.1 DATASETS AND SETUPS

**Datasets.** To evaluate the performance of our model, we conduct experiments based on three commonly used datasets in DDI event prediction task, including ZhangDDI (Zhang et al., 2017), DeepDDI (Ryu et al., 2018) and ChChMiner (Zitnik et al.). Details are present in Appendix E.1.
**Baselines.** In our extensive assessment, our model is compared with eight advanced DDI event prediction methods, all leveraging molecular graphs as input features. The compared methods include DeepDDI (Ryu et al., 2018), SSI-DDI (Nyamabo et al., 2021), CGIB (Lee et al., 2023a), CMRL (Lee et al., 2023c), MDF-SA-DDI (Lin et al., 2022), DSN-DDI (Li et al., 2023), IE-HGNN (Ye & Qian, 2024) and IGIB-ISE (Zhang et al., 2025b). A more detailed description is in Appendix E.2.

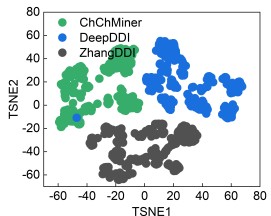

Figure 4: TSNE map for three DDI datasets (3000 drug pairs are respectively selected.)

Table 3: Performance on domain generalization experiments. Bold numbers are the best results, and the standard deviations is in parentheses.

| | ChchMiner | | | DeepDDI | | |
|---|---|---|---|---|---|---|
| | **ACC** (↑) | **AUROC** (↑) | **F1** (↑) | **ACC** (↑) | **AUROC** (↑) | **F1** (↑) |
| DeepDDI | $48.27_{(0.24)}$ | $61.21_{(0.32)}$ | $60.25_{(0.27)}$ | $50.34_{(0.14)}$ | $65.21_{(0.27)}$ | $61.83_{(0.25)}$ |
| SSI-DDI | $51.25_{(0.21)}$ | $60.47_{(0.31)}$ | $62.34_{(0.52)}$ | $53.26_{(0.45)}$ | $67.24_{(0.42)}$ | $63.11_{(0.34)}$ |
| MDF-SA-DDI | $33.54_{(0.12)}$ | $65.34_{(0.32)}$ | $63.55_{(0.54)}$ | $54.63_{(0.34)}$ | $68.50_{(0.25)}$ | $64.17_{(0.21)}$ |
| DSN-DDI | $52.24_{(0.24)}$ | $62.45_{(0.28)}$ | $64.20_{(0.09)}$ | $54.86_{(0.21)}$ | $68.25_{(0.24)}$ | $65.34_{(0.24)}$ |
| CGIB | $55.21_{(0.21)}$ | $67.54_{(0.46)}$ | $64.53_{(0.44)}$ | $55.37_{(0.29)}$ | $68.48_{(0.45)}$ | $65.44_{(0.32)}$ |
| CMRL | $55.76_{(0.21)}$ | $68.14_{(0.23)}$ | $64.82_{(0.15)}$ | $56.43_{(0.55)}$ | $68.45_{(0.21)}$ | $65.62_{(0.45)}$† |
| IE-HGNN | $56.48_{(0.19)}$ | $68.32_{(0.27)}$ | $65.01_{(0.22)}$ | $56.35_{(0.18)}$ | $68.63_{(0.26)}$ | $65.48_{(0.28)}$ |
| IGIB-ISE | $57.06_{(0.17)}$† | $68.63_{(0.24)}$† | $65.11_{(0.19)}$† | $57.21_{(0.21)}$† | $68.67_{(0.23)}$† | $65.57_{(0.25)}$ |
| Ours | $\mathbf{59.25}_{(0.15)}$ | $\mathbf{69.22}_{(0.31)}$ | $\mathbf{65.25}_{(0.26)}$ | $\mathbf{58.12}_{(0.09)}$ | $\mathbf{68.72}_{(0.42)}$ | $\mathbf{65.76}_{(0.23)}$ |

**Metric.** Three metrics are employed to evaluate the model performance: accuracy (ACC), area under the receiver operating characteristic (AUROC), harmonic mean of precision and recall (F1). All experiments are repeated eight times with the same dataset split, and average result is presented.

## 4.2 MODEL PERFORMANCE

Similar to previous studies (Deac et al., 2019; Nyamabo et al., 2021), we first performed the transductive setting that is the common method evaluation scheme, where the entire dataset is randomly split and aims to predict the undiscovered DDI events among known drugs. We split the dataset into training (60%), validation (20%), and test (20%) parts. Key observations can be got:

**Obs.1:** I2Mole exhibits the excited predictive performance in transductive setting. The results of our model and eight baseline models are presented in Table 1. We observe that our model demonstrates the optimal predictive performance across three different scales of datasets. Regarding the ACC evaluation metric, it outperforms other models on the ZhangDDI and DeepDDI datasets, while its performance on the ChChMiner dataset is comparable to IGIB-ISE.

**Obs.2:** I2Mole shows more pronounced performance improvements on the large-scale DeepDDI dataset. The model's performance across different datasets may be influenced by variations in dataset characteristics, where larger datasets imply a greater diversity of drugs and more complex DDI relationships. Compared to the second-best model, I2Mole has improved by 0.98% on the large-scale DeepDDI dataset, while only by 0.45% on the medium and small-scale datasets in AUROC index.

## 4.3 GENERALIZATION TEST

In this section, we evaluated I2Mole's generalizability by inductive settings and domain shifting tests. Type 1 aims to predict potential interaction properties between known and unseen drugs, while Type 2 aims to predict potential interaction properties between unseen and unseen drugs as in Table 2.

**Obs.3:** I2Mole demonstrates excellent generalization ability on inductive settings. We assessed the generalization capability on I2Mole to unseen drugs, which holds significant practical and real-world implications. This process was implemented by partitioning drugs, and the testing results, compared with baseline models, are documented in Table 2. Evidently, when predicting with new drugs, the performance of all models experiences varying degrees of decline. However, I2Mole exhibiting excellent predictive performance has the minimized sensitivity to unseen drug pairs.

**Obs.4:** I2Mole shows robust performance on domain generalization experiments. To investigate the impact of domain shifting on generalization, we transfer a model trained on a smaller dataset to a larger one. Specifically, I2Mole, trained on the ZhangDDI dataset, is tested on two other datasets. Notably, ZhangDDI and DeepDDI exhibit entirely distinct distributions of molecular species, as depicted in Figure 4. I2Mole outperforms other baseline models consistently across all conditions, underscoring its superior generalization capability, as recorded in Table 3.

**Obs.5:** I2Mole demonstrates superior performance in scaffold and size splitting experiments. As presented in Table 20 (Appendix L), our proposed model consistently surpasses state-of-the-art methods, achieving the highest accuracy in both scaffold and size splits. These results underscore the advantages of our approach, enabling the model to effectively extract rationales with impressive generalization capabilities, and perform robustly across different test scenarios.

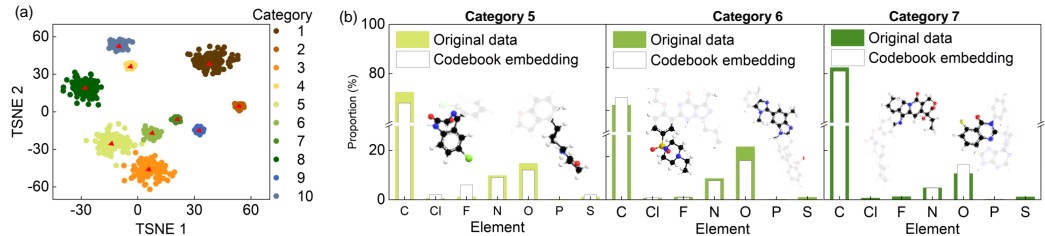

Figure 5: Environmental codebook vectors analysis. (a) TSNE dimensionality reduction plot of drug molecular pairs and the 10-class environment codebook vectors. Different colors represent the chosen codebook vectors, with red dots within clusters indicating the codebook vectors location. (b) Elemental composition of molecular pairs in clusters 5, 6, and 7 (colored) compared to the elemental composition represented by the codebook vectors (blank), along with an example pair of molecules.

Table 4: Ablation experiment. Intermolecular interaction denotes as $\Delta$.

|  | ZhangDDI | | |
| --- | --- | --- | --- |
|  | **ACC** | **AUROC** | **F1** |
| w/o VQ | $74.52_{(0.11)}$ | $83.61_{(0.13)}$ | $74.01_{(0.24)}$ |
| w/o $\Delta$ | $84.51_{(0.22)}$ | $87.21_{(0.27)}$ | $80.21_{(0.33)}$ |
| w/o GIB | $84.72_{(0.08)}$ | $87.21_{(0.24)}$ | $81.07_{(0.43)}$ |
| Ours | $88.64_{(0.24)}$ | $95.12_{(0.12)}$ | $85.87_{(0.20)}$ |

Table 5: Sensitivity analysis for $\beta$ and $\gamma$ (ACC indicator).

|  | ZhangDDI | | | | |
| --- | --- | --- | --- | --- | --- |
|  | **0** | **1E-5** | **1E-4** | **1E-3** | **0.1** |
| $\beta$ | $85.71_{(0.02)}$ | $85.84_{(0.08)}$ | $88.64_{(0.24)}$ | $86.83_{(0.03)}$ | $56.46_{(0.03)}$ |
|  | **2E-5** | **1E-4** | **2E-4** | **5E-4** | **1E-3** |
| $\gamma$ | $88.31_{(0.08)}$ | $88.64_{(0.12)}$ | $88.21_{(0.02)}$ | $88.64_{(0.24)}$ | $88.32_{(0.01)}$ |

### 4.4 EXPLORING THE IMPACT AND EFFECTS OF ENVIRONMENT CODEBOOK

In this section, we provide an intuitive understanding through t-SNE analysis of environment vectors and molecular embeddings from ChchMiner dataset, as presented in Figure 5.

**Obs.6:** Different environment embeddings in the environment codebook have clear boundaries in the visualization results. The 10 distinct environment embeddings exhibit clear distinctions (Figure 5 (a)), ensuring that the model adequately learns different types of environmental variables and thereby enhances its generalization. Moreover, different molecular substructure embeddings are tightly centered around their corresponding environment embedding. This suggests that updating the codebook vector is essentially equivalent to performing clustering on the molecular embeddings, with the environment embeddings serving as the clustering centers as shown in Figure 5 (a).

**Obs.7:** Different environment codes tend to encode the local environments of various molecular pairs. Figure 5 (b) shows the distribution of atom types for each environmental embedding, which is close to real-world data. Notably, significant differences exist between environment codes; for example, carbon is predominant in Category 7, while nitrogen and oxygen play important roles in Categories 5 and 6. These environmental embeddings represent the non-core substructures of molecular pairs. For each codebook category, we provide examples of molecular pairs in Figure 5 (b), illustrating the types of real-world substructures represented. More analysis are in Appendix M and N.

## 5 ABLATION STUDY AND SENSITIVITY ANALYSIS

**Ablation study.** To further investigate the role of each component, we conducted a series of ablation studies. As shown in Table 4, removing the GIB module reduced the model's ability to capture core substructures, limiting its performance. Similarly, the removal of intermolecular interaction disrupted accurate chemical modeling, degrading the model's capabilities. Notably, eliminating VQ module led to substantial performance drops, highlighting the importance of codebook and vector quantization operation (more details in Appendix H and complexity are in Appendix I and J).

**Sensitivity analysis.** We investigate the sensitivity of $\beta$ and $\gamma$, which govern the trade-off between prediction and compression, and the codebook updating process, respectively. These parameters correspond to the weights of $\mathcal{L}_{MI}$ and $\mathcal{L}_{vq}$ in Equation 24. Overall, the model demonstrates robustness to variations in $\beta$ and $\gamma$, but performance degrades significantly when $\beta$ is sharply increased. More detailed sensitivity analysis results,are presented in Appendix G.

## 6 CONCLUSION AND FUTURE OUTLOOK

In this work, we introduce I2Mole, a novel framework for precise DDI prediction that aims to address the imbalance between training and testing data distributions commonly observed in real-world scenarios. I2Mole constructs a merged graph to capture complex molecular interactions and, through an enhanced information bottleneck theory to extract invariant subgraphs. Meanwhile, we design an environment codebook based on the molecular environments, which encodes environmental information and integrates it into data from diverse distributions, further improving the model's generalization capability. I2Mole enables the rapid identification of potential DDIs and reducing risks associated with drug misuse. The limitations of I2Mole are discussed in Appendix K.

## 7 REPRODUCIBILITY

We provide the complete implementation in the repository along with guidance on how to reproduce our results. Our code is available at `https://anonymous.4open.science/r/I2Mol-C616`.

## 8 ETHICS STATEMENT

Our study does not involve human participants, personal data, or sensitive information. The datasets and resources used are either publicly available or released under appropriate licenses. We confirm that our research does not raise any ethical concerns related to privacy, safety, fairness, or potential misuse. The contributions of this work are intended solely for advancing scientific research and are not designed or evaluated for harmful applications.

## 9 ACKNOWLEDGMENT

This paper is partially supported by the National Natural Science Foundation of China (No.12227901). The AI-driven experiments, simulations and model training were performed on the robotic AI-Scientist platform of Chinese Academy of Sciences., Anhui Science Foundation for Distinguished Young Scholars (No.1908085J24), Natural Science Foundation of China (No.62502491).

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

CONTENTS

## A    REGARDING THE USE OF LLMs

In this work, LLMs are used solely for polishing and refining the writing. All substantive content, ideas, and analyses are authored and created by the authors. The LLMs are only employed to improve clarity, grammar, and overall readability, and did not contribute to the generation of scientific content or results.

## B    RELATED WORK

### B.1    DRUG-DRUG INTERACTION (DDI) PREDICTION

In recent years, computational approaches, particularly employing machine learning and deep learning methods, have emerged as indispensable tools for swiftly and economically predicting potential DDIs (Ryall & Tan, 2015; Jaaks et al., 2022). Initially, DDI prediction models predominantly focused on drug attribute information, assuming that similar drugs would exhibit common interactions (Ryu et al., 2018; Deng et al., 2020). For instance, Gottlieb et al. (2012) utilized seven types of drug features to construct similarity vectors, forming a DDI prediction model based on logistic regression. Ferdousi et al. (2017) designed a deep neural network using drug molecular similarity vectors as descriptors for predicting potential DDIs. Recently, there has been a shift towards graph-based DDI prediction methodologies. Zhong et al. (2019) employed Graph Convolutional Neural Networks (GCNNs) for message aggregation and an attention-based pooling method for DDI prediction. Given that the interaction between two drugs is influenced by their specific substructures and functions, recent efforts have focused on substructure extraction and interaction Harrold & Zavod (2014); Fu et al. (2020). For instance, Yu et al. (2022a) utilized functional group information of drug molecules as their substructures, while Nyamabo et al. (2021) introduced the Substructure-Substructure Interaction for Drug-Drug Interaction (SSI-DDI) method, employing graph attention network (GAT) layers to extract substructure representations and co-attention layers to model interactions among substructures.

Despite the proficiency of existing methodologies in elucidating essential structural characteristics of individual molecular models, considerable variation in crucial substructures may occur during molecular interactions (Tang et al., 2023; Lee et al., 2023c). Notably, while some pioneering work like DSIL-DDI (Tang et al., 2023) and CMRL (Lee et al., 2023c) has provided foundational insights, a noticeable gap remains in comprehensively modeling intermolecular interactions. CMRL (Lee et al., 2023c) innovatively incorporates conditional graph information bottleneck theory to obtain rationales, simultaneously considering a second drug as a conditional factor during drug subgraph generation (Lee et al., 2023b). However, prevailing methodologies encounter limitations in adequately capturing molecular interactions, particularly at the atomic level. Moreover, integrating a comprehensive profile of interacting molecules into subgraph generation poses significant challenges, including overwhelming complexity and the risk of incorporating redundant information (Jia et al., 2009).

### B.2    OOD GENERALIZATION

The susceptibility of deep neural networks to significant performance degradation under distribution shifts has spurred extensive research on out-of-distribution (OOD) generalization. In response, the invariant rationalization theory has been introduced, aiming to achieve an invariant representation across diverse environments (Chang et al., 2020; Rojas-Carulla et al., 2018b). This theory involves a rationalization module that discerns a crucial subset within the input graph, referred to as rationale, essential for prediction (Ying et al., 2019; Luo et al., 2020). Subsequently, through invariant learning, these rationales are exposed to diverse environments, thereby fortifying the learned representation against environmental fluctuations and effectively bolstering the model's OOD capacity. 1) **sufficiency:** shows sufficient predictive power for the target, 2) **invariance:** contributes to equal (optimal) performance for the downstream tasks across all environments. Certain methods in computer vision Lv et al. (2022); Zhang et al. (2020); Wang et al. (2020) achieve OOD generalization by learning domain-invariant representations. Additionally, methods such as Shen et al. (2018); He et al. (2021); Shen et al. (2020) aim to achieve OOD generalization by decorrelating correlated and irrelevant features, considering the statistical correlation between these features as a major factor for distribution shifts. In terms of molecular applications, DIR (Wu et al., 2022d) introduces an inventive method to unveil invariant rationales by intervening in the training distribution, generating multiple

interventional distributions, and identifying causal rationales consistent across varied distributions. Similarly, MoleOOD (Yang et al., 2022a) suggests that leveraging causal data-generating invariance from substructures across environments, linked to specific properties, holds promise for enhancing OOD generalization. However, learning a domain-invariant representation for intermolecular interaction remains an open problem, and current discussions on OOD issues are limited.

## C PROOFS

### C.1 PROOF OF $\mathcal{L}_{pre}$

*Proof.* Regarding $I(Y; \widetilde{\mathcal{G}})$, we consider $P_\theta\left(Y \mid \widetilde{\mathcal{G}}\right)$ as the variational estimation of $P\left(Y \mid \widetilde{\mathcal{G}}\right)$. Therefore, we can proceed with the following derivation:

$$
\begin{aligned}
I(Y; \widetilde{\mathcal{G}}) &= \mathbb{E}_{(Y,\widetilde{\mathcal{G}})} \log \left[ \frac{P\left(Y \mid \widetilde{\mathcal{G}}\right)}{P(Y)} \right] \\
&= \mathbb{E}_{(Y,\widetilde{\mathcal{G}})} \log \left[ \frac{P_\theta\left(Y \mid \widetilde{\mathcal{G}}\right)}{P(Y)} \right] + \\
&\quad \mathbb{E}_{\widetilde{\mathcal{G}}} \log \left[ KL\left( P\left(Y \mid \widetilde{\mathcal{G}}\right) \| P_\theta\left(Y \mid \widetilde{\mathcal{G}}\right) \right) \right].
\end{aligned}
\tag{25}
$$

Considering the non-negativity property of the Kullback-Leibler divergence, we can conclude that:

$$
\begin{aligned}
I(Y; \widetilde{\mathcal{G}}) &\geq \mathbb{E}_{(Y,\widetilde{\mathcal{G}})} \log \left[ \frac{P_\theta\left(Y \mid \widetilde{\mathcal{G}}\right)}{P(Y)} \right] \\
&= \mathbb{E}_{(Y,\widetilde{\mathcal{G}})} \log \left[ P_\theta\left(Y \mid \widetilde{\mathcal{G}}\right) \right] + H(Y).
\end{aligned}
\tag{26}
$$

As $H(Y)$ remains constant across all data, it can be omitted, resulting in the final formulation of this term:

$$
\mathcal{L}_{pre} := \mathbb{E}_{(Y,\widetilde{\mathcal{G}})} \log \left[ P_\theta\left(Y \mid \widetilde{\mathcal{G}}\right) \right].
\tag{27}
$$

$\square$

### C.2 PROOF OF $\mathcal{L}_{\mathrm{MI}}$

*Proof.* We first use a readout function to obtain the graph representation $z_{\widetilde{G}_{\mathrm{IB}}}$ of the perturbed graph $\widetilde{G}_{\mathrm{IB}}$. And we assume these is no information loss in this process. Therefore we have $I\left(z_{\widetilde{G}_{\mathrm{IB}}}; \widetilde{G}\right) \approx$

$I\left(\widetilde{G}_{\mathrm{IB}};\widetilde{G}\right)$. Now we bound $I\left(z_{\widetilde{G}_{\mathrm{IB}}};\widetilde{G}\right)$ using variational approximation:

$$
\begin{aligned}
I\left(z_{\widetilde{G}_{\mathrm{IB}}};\widetilde{G}\right) &= \iint p\left(z_{\widetilde{G}_{\mathrm{IB}}},\widetilde{G}\right)\log\frac{p\left(z_{\widetilde{G}_{\mathrm{IB}}}\mid\widetilde{G}\right)}{p\left(z_{\widetilde{G}_{\mathrm{IB}}}\right)}\,\mathrm{d}z_{\widetilde{G}_{\mathrm{IB}}}\,\mathrm{d}\widetilde{G} \\
&= \iint p\left(z_{\widetilde{G}_{\mathrm{IB}}},\widetilde{G}\right)\log\frac{p\left(z_{\widetilde{G}_{\mathrm{IB}}}\mid\widetilde{G}\right)}{q\left(z_{\widetilde{G}_{\mathrm{IB}}}\right)}\,\mathrm{d}z_{\widetilde{G}_{\mathrm{IB}}}\,\mathrm{d}\widetilde{G} \\
&\quad + \iint p\left(z_{\widetilde{G}_{\mathrm{IB}}},\widetilde{G}\right)\log\frac{q\left(z_{\widetilde{G}_{\mathrm{IB}}}\right)}{p\left(z_{\widetilde{G}_{\mathrm{IB}}}\right)}\,\mathrm{d}z_{\widetilde{G}_{\mathrm{IB}}}\,\mathrm{d}\widetilde{G} \\
&= \mathbb{E}_{p(\widetilde{G})}\left[\mathrm{KL}\left(p\left(z_{\widetilde{G}_{\mathrm{IB}}}\mid\widetilde{G}\right)\|\,q\left(z_{\widetilde{G}_{\mathrm{IB}}}\right)\right)\right] \\
&\quad - \mathbb{E}_{p\left(z_{\widetilde{G}_{\mathrm{IB}}}\mid\widetilde{G}\right)}\left[\mathrm{KL}\left(p\left(z_{\widetilde{G}_{\mathrm{IB}}}\right)\|\,q\left(z_{\widetilde{G}_{\mathrm{IB}}}\right)\right)\right] \\
&\leq \mathbb{E}_{p(\widetilde{G})}\left[\mathrm{KL}\left(p\left(z_{\widetilde{G}_{\mathrm{IB}}}\mid\widetilde{G}\right)\|\,q\left(z_{\widetilde{G}_{\mathrm{IB}}}\right)\right)\right],
\end{aligned}
\tag{28}
$$

where $q\left(z_{\widetilde{G}_{\mathrm{IB}}}\right)$ is the variational approximation to $p\left(z_{\widetilde{G}_{\mathrm{IB}}}\right)$. And the inequality is due to the fact that Kullback-Leibler divergence is non-negative. We assume that $q\left(z_{\widetilde{G}_{\mathrm{IB}}}\right)$ is a noninformative distribution following VIB Alemi et al. (2016). That is, we obtain $q\left(z_{\widetilde{G}_{\mathrm{IB}}}\right)$ by aggregating the node representations in a fully perturbed graph. The noise $\epsilon_{\widetilde{G}}\sim\mathcal{N}\left(\mu_h,\sigma_h^2\right)$ is sampled from the Gaussian distribution. $\mu_h,\sigma_h^2$ are mean and variance of $h_j$ in $\widetilde{G}$.

When we choose sum pooling as the readout function, we have:

$$
q\left(z_{\widetilde{G}_{\mathrm{IB}}}\right) = \mathcal{N}\left(m_{\widetilde{G}}\mu_h, m_{\widetilde{G}}\sigma_h^2\right).
\tag{29}
$$

This is because the summation of Gaussian distributions is also a Gaussian distribution. Then, for $p\left(z_{\widetilde{G}_{\mathrm{IB}}}\mid\widetilde{G}\right)$, we have:

$$
\begin{aligned}
&p\left(z_{\widetilde{G}_{\mathrm{IB}}}\mid\widetilde{G}\right) \\
&= \mathcal{N}\left(m_{\widetilde{G}}\mu_h + \sum_{j=1}^{m_{\widetilde{G}}}\lambda_j h_j - \sum_{j=1}^{m_{\widetilde{G}}}\mu_h\lambda_j, \sum_{j=1}^{m_{\widetilde{G}}}(1-\lambda_j)^2\sigma_h^2\right).
\end{aligned}
\tag{30}
$$

Plug Equation 29 and Equation 30 into Equation 28 and we have:

$$
\begin{aligned}
&I\left(z_{\widetilde{G}_{\mathrm{IB}}};\widetilde{G}\right) \\
&\leq \int p(\widetilde{G})\left(-\frac{1}{2}\log A_{\widetilde{G}} + \frac{1}{2m_{\widetilde{G}}}A_{\widetilde{G}} + \frac{1}{2m_{\widetilde{G}}}B_{\widetilde{G}}^2\right)\mathrm{d}\widetilde{G} \\
&\quad + \int\frac{1}{2}p(\widetilde{G})\log m_{\widetilde{G}}\,\mathrm{d}\widetilde{G} \\
&= \int p(\widetilde{G})\left(-\frac{1}{2}\log A_{\widetilde{G}} + \frac{1}{2m_{\widetilde{G}}}A_{\widetilde{G}} + \frac{1}{2m_{\widetilde{G}}}B_{\widetilde{G}}^2\right)\mathrm{d}\widetilde{G} + C,
\end{aligned}
\tag{31}
$$

where $A_{\widetilde{G}} = \sum_{j=1}^{m_{\widetilde{G}}}(1-\lambda_j)^2$ and $B_{\widetilde{G}} = \frac{\sum_{j=1}^{m_{\widetilde{G}}}\lambda_j(h_j-\mu_h)}{\sigma_h}$. $C$ is a constant and can be ignored in the optimization process.

$\square$

# D  THE DETAILED FEATURES FOR ATOMS, BONDS AND MOLECULAR GLOBAL

A comprehensive overview of the selected atom, bond, and global input features is presented in Table 6. The initial step involves the conversion of the SMILES string of both solute and solvent into a graph structure using the RDKit package. This package is employed not only for graph creation but also for the computation of atom and bond features for each graph. The selection of features was restricted to those computable in RDKit to mitigate the computational expenses associated with performing quantum mechanics calculations for the entire dataset. In order to standardize the lengths of the bond, atom, and global feature vectors, a linear transformation is applied to each vector before the commencement of the message-passing steps.

Table 6: Atoms (nodes), bonds (edges), and global features for molecular representation

| Atomic features ($\mathcal{V}$) | Bond features ($\mathcal{E}$) | Global features ($\mathcal{U}$) |
|---|---|---|
| Atomic species | Bond type | Total No. of atoms |
| No. of bonds | Conjugated status | Total No. of bonds |
| No. of bonded H atoms | Ring size | Molecular weight |
| Ring status | Stereo-chemistry | – |
| Valence | – | – |
| Aromatic status | – | – |
| Hybridization type | – | – |
| Acceptor status | – | – |
| Donor status | – | – |
| Partial charge | – | – |

# E  EXPERIMENTAL SETTINGS

In this section, we will provide a comprehensive overview of our experimental setup. Section E.1 will provide detailed information about all the datasets utilized in the experiments. Subsequently, Section E.2 will offer a basic introduction to the baseline methods incorporated in our study. Following that, Section E.3 will delineate the diverse hyperparameters employed in the network architecture of our model. Additionally, it will elucidate the search space for hyperparameters and present the optimal hyperparameters.

## E.1  DATASETS

- **ZhangDDI** Zhang et al. (2017) is a small-scale dataset, including 548 drugs with 48,548 pairwise interaction data points, encompassing various types of similarity information for these drug pairs.

- **ChChMiner** Zitnik et al. a medium-scale dataset, comprises 1,514 drugs and 48,514 labeled DDIs, sourced from drug labels and scientific publications.

- **DeepDDI** Ryu et al. (2018) is a larger-scale dataset with 1,704 drugs and 192,284 labeled DDIs, along with comprehensive side-effect information.

These datasets provide detailed drug information, including SMILES string representations, forming a robust foundation for evaluating the proposed model.

## E.2  BASELINES

In this chapter, we will provide a brief introduction to the baseline models mentioned in the experimental section. In our extensive assessment, our model is compared with eight advanced DDI event prediction methods, all leveraging molecular graphs as input features.

**DeepDDI.** Ryu et al. (2018)It is based on the structural similarity profile between input drugs and others.

**SSI-DDI.** Nyamabo et al. (2021) it use a 4-layer GAT network to extract substructures at different levels, and finally complete the final prediction based on the co-attention mechanism.

**CGIB.** Lee et al. (2023b) Based on the graph conditional information bottleneck theory, conditional subgraphs are extracted to complete the interaction between molecules.

**CMRL.** Lee et al. (2023c) it detects the core substructure that is causally related to chemical reactions. we introduce a novel conditional intervention framework whose intervention is conditioned on the paired molecule. With the conditional intervention framework.

**MDF-SA-DDI.** Lin et al. (2022), achieving DDI prediction by incorporating multi-source drug fusion, multi-source feature fusion and transformer self-attention mechanism.

**DSN-DDI.** Li et al. (2023) it employs local and global representation learning modules iteratively and learns drug substructures from the single drug ('intra-view') and the drug pair ('inter-view') simultaneously.

**IE-HGNN.** Ye & Qian (2024) It introduces an internal–external bi-view hypergraph neural network, where cross-interaction message passing is applied to capture molecular relational patterns and reduce edge redundancy in paired molecular graphs.

**IGIB-ISE.** Zhang et al. (2025b) It integrates an iterative substructure extraction framework with the Interactive Graph Information Bottleneck, progressively refining interactive core substructures between drug pairs to improve accuracy and interpretability in molecular relational learning.

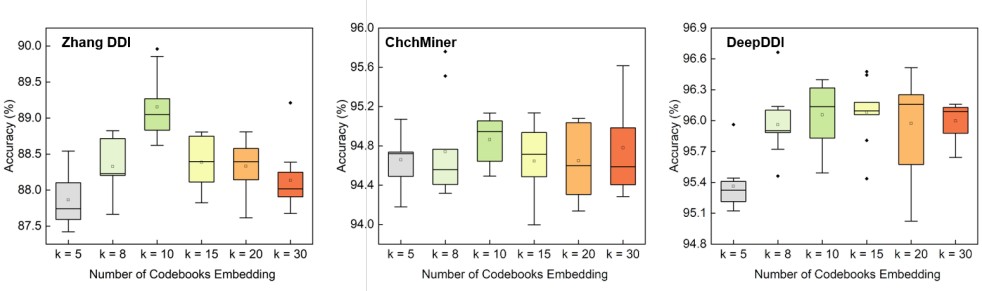

Figure 6: Test results of varying numbers of environmental vectors in the environment codebook in a transductive setting.

### E.3    PARAMETER SETTING

**Model architecture.** For intramolecular message passing, we employ a 3-layer Gated Graph Convolutional Network (GatedConv). For intermolecular message passing, we utilize a 3-layer Graph Attention Network (GAT). As for the pooling layer, we opt for the set2set network. The detailed hyperparameters are present in Table 7.

**Model Training.** The model is trained using the Adam optimizer Kingma & Ba (2014) with an initial learning rate of $1 \times 10^{-4}$, which is increased to 0.5, employing a batch size of 32. Binary cross-entropy loss (BCE) is utilized as the training loss function. Training is terminated if the validation error does not decrease for 150 epochs or if the maximum training limit of 300 epochs is reached. I2Mole is implemented in the PyTorch framework and executed on Tesla A100 40GB hardware.

## F    RESULTS SIGNIFICANCE ANALYSIS

In the transductive setting, which is a conventional testing method where the dataset is randomly divided into training, validation, and test sets, we repeated the experiment 8 times and calculated the mean, variance, and p-value of the ACC, as shown in the Table 8, compared to the second-best model, I2Mole improved by 0.98% on the large-scale DeepDDI dataset, with an average improvement of approximately 0.05. The p-values are less than 0.05, proving that the improvements of our model are

Table 7: Hyperparameter specifications.

| Network layer hyperparameters | | | | | | |
|---|---|---|---|---|---|---|
| **GatedConv** | | **GAT** | | | **FC** | |
| Num-layers | **3**, 4, 5 | Num-layers | **3**, 4, 5 | | Num-layers | 3, **4**, 5 |
| Hidden-size | 200, **400**, 600, 800 | Hidden-size | 200, **400**, 600, 800 | | Dropout | 0.5 |
| Layers | **3**, 4, 5 | Layers | **3**, 4, 5 | | Hidden-size | 300, **400**, 500, 600 |
| Activation | `LeakyReLU` | Activation | `LeakyReLU` | | | |
| Training hyperparameters | | | | | | |
| Batch-size | 32 | Learning rate | **0.0001**, 0.0005, 0.01, 0.005 | | $\beta$ | 0, $1e^{-5}$, **$1e^{-4}$**, **$1e^{-3}$**, $1e^{-1}$ |
| Environment codebook hyperparameters | | | | | | |
| Num of environment | 5, 8, **10**, 15, 20, 30 | $\gamma$ | $2e^{-5}$, $1e^{-4}$, $2e^{-4}$, $5e^{-4}$, $1e^{-3}$ | | | |

statistically significant. Therefore, although this is a limited improvement, the results are substantial and significant.

In the inductive setting, which is a commonly used generalization testing method, a molecule (type 1) or a molecule pair (type 2) is removed in the test set to verify the model's generalization ability. Notably, in the generalization tests, I2Mole achieved an average improvement of 1.05% in type 1 scenarios and 1.20% in type 2 scenarios. This demonstrates I2Mole's ability to generalize to unseen molecules.

In domain generalization experiments, which is a challenging testing method, we trained and tested on datasets from different domains to verify I2Mole's generalization ability. Clearly, I2Mole achieved the best generalization results, with an average improvement of 2.71% (Acc index), significantly outperforming the improvements in transductive and inductive settings.

Table 8: Significant difference analysis.

| | **Model Performance** | | |
|---|---|---|---|
| | **ZhangDDI** | **ChchMiner** | **DeepDDI** |
| **Second-best model** | $88.08_{(0.26)}$ | $94.92_{(0.21)}$ | $95.85_{(0.19)}$ |
| **Our model** | $88.64_{(0.24)}$ | $95.34_{(0.19)}$ | $96.51_{(0.14)}$ |
| **P-value** | 3.52E-04 | 2.13E-03 | 8.74E-05 |

# G SENSITIVITY ANALYSIS

We conduct an in-depth investigation into the effect of varying the number of environment embeddings on our model's performance, as depicted in Figure 6. The results demonstrate that altering the number of environment embeddings has a negligible impact on test performance across three different-sized test datasets, underscoring the robustness of our model. The optimal performance is achieved when the number of codebook vectors is set to 10, which we have adopted for our final model configuration.

Furthermore, we investigated the impact of varying the proportion of retained relational edges during inter-molecular message passing on the model's performance, as illustrated in Table 9. The results indicate that gradually increasing the proportion of retained relational edges enhances the model's performance. However, beyond a certain threshold, further increments lead to a noticeable decline in performance. Consequently, we selected 20% as the optimal parameter for the proportion of retained relational edges.

Table 9: Sensitivity analysis for retained relational edges ratio.

| | **ZhangDDI** | | |
|---|---|---|---|
| | **ACC** | **AUROC** | **F1** |
| Top_s = 5% | $88.52_{(0.08)}$ | $95.02_{(0.02)}$ | $85.75_{(0.06)}$ |
| Top_s = 10% | $88.60_{(0.10)}$ | $95.10_{(0.03)}$ | $85.80_{(0.20)}$ |
| Top_s = 20% | $88.64_{(0.24)}$ | $95.12_{(0.12)}$ | $85.87_{(0.20)}$ |
| Top_s = 30% | $88.54_{(0.12)}$ | $95.09_{(0.12)}$ | $85.53_{(0.13)}$ |
| Top_s = 50% | $88.34_{(0.14)}$ | $94.87_{(0.10)}$ | $85.22_{(0.11)}$ |

A crucial parameter in this context is the number of environmental samples, denoted as $\theta$. Increasing the number of sampled environments expands the range of simulated molecular interactions under various conditions, though it also introduces additional training overhead. To identify the optimal value of $\theta$, we systematically evaluated the impact of different sampling quantities, ranging from 2 to

6, within the framework of an environment codebook size of $\mathbf{k} = 10$, as shown in Table 10. Based on the results, we have determined the optimal value of $\theta$ to be 4.

Table 10: Sensitivity analysis for the sampling numbers of environmental embedding.

| | ZhangDDI | | | ChchMiner | | | DeepDDI | | |
|---|---|---|---|---|---|---|---|---|---|
| | ACC (↑) | AUROC (↑) | F1 (↑) | ACC (↑) | AUROC (↑) | F1 (↑) | ACC (↑) | AUROC (↑) | F1 (↑) |
| $\theta = 2$ | $88.12_{(0.11)}$ | $94.79_{(0.14)}$ | $85.03_{(0.06)}$ | $94.67_{(0.14)}$ | $98.61_{(0.21)}$ | $95.83_{(0.11)}$ | $96.48_{(0.11)}$ | $97.11_{(0.10)}$ | $97.27_{(0.06)}$ |
| $\theta = 3$ | $88.17_{(0.15)}$ | $94.74_{(0.10)}$ | $85.07_{(0.20)}$ | $94.43_{(0.14)}$ | $98.68_{(0.07)}$ | $95.71_{(0.08)}$ | $96.23_{(0.10)}$ | $97.93_{(0.18)}$ | $97.37_{(0.10)}$ |
| $\theta = 4$ | $88.64_{(0.24)}$ | $95.12_{(0.12)}$ | $85.87_{(0.20)}$ | $95.34_{(0.19)}$ | $98.84_{(0.10)}$ | $96.21_{(0.25)}$ | $96.51_{(0.14)}$ | $99.04_{(0.22)}$ | $97.53_{(0.16)}$ |
| $\theta = 5$ | $88.29_{(0.13)}$ | $94.87_{(0.13)}$ | $85.41_{(0.12)}$ | $94.38_{(0.19)}$ | $98.82_{(0.18)}$ | $95.61_{(0.10)}$ | $96.18_{(0.12)}$ | $98.89_{(0.12)}$ | $97.28_{(0.08)}$ |
| $\theta = 6$ | $88.24_{(0.14)}$ | $94.85_{(0.15)}$ | $85.09_{(0.10)}$ | $94.57_{(0.16)}$ | $98.76_{(0.10)}$ | $95.78_{(0.08)}$ | $96.34_{(0.12)}$ | $97.00_{(0.13)}$ | $97.37_{(0.11)}$ |

## H  VQ MODULE ANALYSIS

To further illustrate the role of the VQ module, we introduced extra two variants for comparison: (1) RD Noise Variant: In this version, noise in the environment codebook is entirely random, mimicking the effects of random noise injection. (2) Instance-Dependent (ID) Noise Variant: Here, we sampled new environments from the environment codebook and added them as small perturbations to the instance-dependent environment. The Gaussian distribution of this noise is determined by the mean and variance of subgraph node vectors, emphasizing instance-dependent noisy perturbations.

Table 11: Performance comparison across different DDI datasets.

| | ZhangDDI | | ChchMiner | | DeepDDI | |
|---|---|---|---|---|---|---|
| Method | Acc (↑) | AUROC (↑) | Acc (↑) | AUROC (↑) | Acc (↑) | AUROC (↑) |
| RD noise | $87.21_{(0.11)}$ | $93.76_{(0.13)}$ | $93.47_{(0.08)}$ | $97.52_{(0.07)}$ | $92.39_{(0.38)}$ | $97.01_{(0.39)}$ |
| ID noise | $88.02_{(0.06)}$ | $94.47_{(0.08)}$ | $94.27_{(0.12)}$ | $98.54_{(0.06)}$ | $94.56_{(0.10)}$ | $97.42_{(0.31)}$ |
| Ours | $88.64_{(0.24)}$ | $95.12_{(0.12)}$ | $95.34_{(0.19)}$ | $98.84_{(0.10)}$ | $96.51_{(0.14)}$ | $99.04_{(0.22)}$ |

Our experimental results demonstrate the superiority of the VQ module and the proposed optimization strategy, particularly in improving the model's robustness and ability to generalize across diverse chemical environments.

## I  DATA SCALABILITY ANALYSIS

We have included the time and space complexity results for our model and various baseline models, along with a comparison of model parameters in Table 12 and Table 14 . This table clearly showcases the results of various model parameters, time and computational complexity. Compared to other baseline models, I2Mole exhibits a significantly larger number of total parameters, leading to substantially higher time consumption and computational complexity than the other baselines.

As the amount of training data increases and the test data decreases, the model's performance exhibits a notable improvement. It is particularly worth mentioning that the model shows significant gains during the initial stages, but further increasing the training data yields only marginal improvements. Further, we evaluated the model performance under different training data sizes and model parameter conditions in Table 13. We control the model parameters by adjusting the number of message-passing layers, dimensions of embeddings and feedforward NN.

I2Mole also could naturally generalize to other molecular relational learning tasks due to its pairwise merged-graph design and interaction-focused information bottleneck theory. Therefore, we conducted an additional experiment to substantiate the model's applicability beyond classification-only DDI tasks. Specifically, we adapted I2Mole for regression by (1) replacing the final classification head with a linear regression layer, (2) removing activation, normalization, and dropout, and (3) switching the loss function from BCEWithLogitsLoss to MSELoss. Without further hyperparameter tuning, we evaluated the model on five standard solute–solvent datasets involving Gibbs free energy of solvation and hydration free energy (experimental and calculated) (Du et al., 2024; 2025a). Despite minimal adaptation, I2Mole achieves competitive performance, comparable to strong baselines as present in Table 15.

## J   COMPUTATIONAL COMPLEXITY

Alongside increasing the model's complexity, there is a clear rise in computational time, accompanied by performance improvements. This could be attributed to the more intricate models being able to capture deeper inter-molecular relationships, thereby enhancing performance. However, when the model's parameters are further increased, its performance starts to degrade. This decline is likely due to overfitting, as the model becomes overly complex, leading to difficulties in convergence during training.

Table 12: Computational complexity analysis on I2Mole on ZhangDDI dataset.

| Model Parameter (M) | 23.4 | 26.8 | 29.5 | 31.7 | 35.4 | 38.4 | 40.3 | 44.5 | 49 |
|---|---|---|---|---|---|---|---|---|---|
| Time Consumption (h) | 7.4 | 8.3 | 9.5 | 10.4 | 12.17 | 13.23 | 15.6 | 18.4 | 22.3 |
| Performance (ACC) | 83.61 | 85.94 | 87.43 | 88.27 | 88.64 | 88.64 | 88.35 | 87.91 | 86.87 |

Table 13: Data scalability analysis on I2Mole on ZhangDDI dataset.

| Training Data Ratio (%) | 10 | 20 | 30 | 40 | 50 | 60 | 70 | 80 | 90 |
|---|---|---|---|---|---|---|---|---|---|
| Test Data Ratio (%) | 45 | 40 | 35 | 30 | 25 | 20 | 15 | 10 | 5 |
| Performance (ACC) | 53.27 | 65.27 | 72.34 | 80.34 | 88.27 | 88.64 | 89.01 | 91.34 | 92.25 |

The higher computational cost of I2Mole mainly arises from edge-level aggregation and message passing, as the constructed merged graph contains more relational edges, which are inherent to graph neural networks. We report two effective strategies that reduce cost while preserving most accuracy.

- Improving the message-passing mechanism by using lightweight modules in certain layers. As shown in the Table 16, replacing the current MPNN layer with a standard GIN backbone reduces the complexity to approximately one-fifth of the original, with only ∼4% drop in performance. Therefore, we can consider replacing certain layers with GIN to reduce model complexity while ensuring performance.

- By constructing a Molecular Merged Hypergraph Neural Network Du et al. (2025b) , specific substructures of molecules, such as functional groups, are defined as hypernodes, reducing the number of nodes in the merged graph and thus lowering the model complexity. We have made preliminary attempts with the Hypergraph Neural Network (further optimizations is not within the scope of this work.), where the computational time can be reduced to approximately half of the original, and for larger molecules, the reduction in time consumption is even more significant, without sacrificing much predictive accuracy.

- For further exploration, we consider replacing the current MPNN layers (Table 18, using 3 layers) with GIN layers to better balance model complexity and performance. Our findings show that substituting two of the three layers with GIN significantly reduces model complexity (approximately 25%) while resulting in only a marginal performance drop of ∼0.36%. Despite this slight decrease, the model still achieves SOTA compared to the baseline. In addition, integrating the model with a hypergraph neural network framework—by predefining functional groups or motifs as hypernodes—can further reduce model complexity as present in Table 18. However, we observe that this strategy leads to a more noticeable decline in performance, about ∼3.15%. Therefore, employing lightweight GNN backbones and hypergraph frameworks to improve message aggregation and reduce the number of atomic nodes presents a highly promising method, but further optimization and parameter tuning are needed.

## K   LIMITATION ANALYSIS

I2Mole, based on the drug pair merged graph, achieves the extraction of rationale subgraphs in molecular interactions and combines the trained environment codebook, significantly enhancing generalization capabilities in domain generalization experiments. However, considering the rapid advancements in the pharmaceutical field and real-world prescription scenarios, we foresee improvements to the current framework in three key aspects:

Table 14: Comparison of ZhangDDI, ChchMiner, and DeepDDI across different models.

| Model | Metric | ZhangDDI | ChchMiner | DeepDDI |
|---|---|---|---|---|
| CGIB (Lee et al., 2023b) | ACC | 87.32 | 94.37 | 95.76 |
| | Time (h) | 1.5 | 0.59 | 3.73 |
| | Memory (G) | 5.1 | 3.9 | 7.4 |
| | Parameters (M) | 11 | 11 | 11 |
| CRML (Lee et al., 2023c) | ACC | 87.78 | 94.43 | 95.49 |
| | Time (h) | 1.3 | 0.47 | 3.24 |
| | Memory (G) | 4 | 3.4 | 6.1 |
| | Parameters (M) | 10 | 10 | 10 |
| SSI-DDI (Nyamabo et al., 2021) | ACC | 86.97 | 93.26 | 94.27 |
| | Time (h) | 1.77 | 0.65 | 4.08 |
| | Memory (G) | 3.1 | 2.7 | 4.4 |
| | Parameters (M) | 13 | 13 | 13 |
| DSN-DDI (Li et al., 2023) | ACC | 87.65 | 94.25 | 95.74 |
| | Time (h) | 1.2 | 0.43 | 3.08 |
| | Memory (G) | 2.9 | 3.6 | 4.1 |
| | Parameters (M) | 0.19 | 0.19 | 0.19 |
| IGIB-ISE (Zhang et al., 2025b) | ACC | 88.08 | 94.92 | 95.85 |
| | Time (h) | 8.7 | 2.9 | 22.8 |
| | Memory (G) | 36.2 | 27.3 | 39.4 |
| | Parameters (M) | 10.5 | 10.5 | 10.5 |
| MMGNN (Du et al., 2024) | ACC | 85.40 | 94.18 | 95.12 |
| | Time (h) | 16.8 | 6.1 | 41.2 |
| | Memory (G) | 38.1 | 30.4 | 39.7 |
| | Parameters (M) | 389.10 | 389.10 | 389.10 |
| Explainable GNN (Low et al., 2022b) | ACC | 84.24 | 93.62 | 94.71 |
| | Time (h) | 11.3 | 4.4 | 29.1 |
| | Memory (G) | 20.1 | 15.3 | 18.6 |
| | Parameters (M) | 39.10 | 39.10 | 39.10 |
| MMHNN (Du et al., 2025a) | ACC | 86.17 | 94.55 | 95.43 |
| | Time (h) | 9.6 | 3.8 | 25.6 |
| | Memory (G) | 15.4 | 12.1 | 14.9 |
| | Parameters (M) | 32.26 | 32.26 | 32.26 |
| CasualIB (Zhang et al., 2025a) | ACC | 88.14 | 94.94 | 95.86 |
| | Time (h) | 15.4 | 5.4 | 33.8 |
| | Memory (G) | 22.7 | 17.0 | 20.3 |
| | Parameters (M) | 38.17 | 38.17 | 38.17 |
| I2Mole (Ours) | ACC | **88.64** | **95.34** | **96.51** |
| | Time (h) | 13.23 | 4.9 | 33.07 |
| | Memory (G) | 17 | 13.3 | 11.7 |
| | Parameters (M) | 35.4 | 35.4 | 35.4 |

- We aim to acquire more comprehensive data on drug interaction processes and analyses between molecules, addressing the limitations of current research. In practice, patients often have multiple comorbidities requiring the concurrent use of various drug categories. Thus, the interaction system of multiple drugs remains a critical research area.

- Constructing relationship edges in pairs, as previously done, is an effective strategy for predicting properties between molecular pairs. However, this approach significantly increases the graph's complexity, especially for large, intricate molecules, due to the substantial rise in degrees of freedom, leading to higher computational resource and time consumption.

Table 15: Test performance of different methods across eight independent runs. Mean values are reported, with standard deviations shown in parentheses. (The best result in each column is underlined, while the top-performing baseline is marked with a superscript dagger.)

| | MAE ($\downarrow$) | | | | | RMSE ($\downarrow$) | | | | |
|---|---|---|---|---|---|---|---|---|---|---|
| | FreeSolv | CompSol | Abraham | CompSolv-Exp | MNSol | FreeSolv | CompSol | Abraham | CompSolv-Exp | MNSol |
| D-MPNN | $0.684_{(0.052)}$ | $0.179_{(0.013)}$ | $0.454_{(0.036)}$ | $0.442_{(0.022)}$ | $0.459_{(0.032)}$ | $1.164_{(0.055)}$ | $0.343_{(0.017)}$ | $0.624_{(0.024)}$ | $0.672_{(0.051)}$ | $0.667_{(0.017)}$ |
| Explainable GNN | $0.724_{(0.031)}$ | $0.184_{(0.012)}$ | $0.486_{(0.042)}$ | $0.321_{(0.013)}$ | $0.396_{(0.011)}$ | $1.276_{(0.045)}$ | $0.367_{(0.012)}$ | $0.776_{(0.035)}$ | $0.404_{(0.054)}$ | $0.673_{(0.024)}$ |
| SolvBERT | $0.588_{(0.034)}$ | $0.167_{(0.014)}$ | $0.467_{(0.034)}$ | $0.382_{(0.023)}$ | $0.354_{(0.021)}$ | $1.021_{(0.043)}$ | $0.328_{(0.020)}$ | $0.652_{(0.022)}$ | $0.472_{(0.041)}$ | $0.623_{(0.104)}$ |
| GAT | $0.675_{(0.033)}$ | $0.187_{(0.011)}$ | $0.457_{(0.043)}$ | $0.970_{(0.031)}$ | $0.514_{(0.043)}$ | $1.185_{(0.075)}$ | $0.390_{(0.012)}$ | $0.726_{(0.040)}$ | $0.810_{(0.101)}$ | $0.812_{(0.124)}$ |
| GROVER | $0.623_{(0.054)}$ | $0.155_{(0.022)}$ | $0.307_{(0.035)}$ | $0.382_{(0.023)}$ | $0.354_{(0.024)}$ | $1.015_{(0.022)}$ | $0.332_{(0.016)}$ | $0.475_{(0.044)}$ | $0.491_{(0.053)}$ | $0.672_{(0.027)}$ |
| SMD | $0.574_{(0.036)}$ | $0.162_{(0.014)}$ | $0.374_{(0.024)}$ | $0.633_{(0.044)}$ | $0.427_{(0.034)}$ | $1.113_{(0.015)}$ | $0.317_{(0.011)}$ | $0.516_{(0.065)}$ | $1.023_{(0.152)}$ | $0.682_{(0.032)}$ |
| Uni-Mol | $0.565_{(0.038)}$ | $0.164_{(0.027)}$ | $0.322_{(0.071)}$ | $0.214_{(0.022)}$ | $0.374_{(0.021)}$ | $1.002_{(0.064)}$ | $0.303_{(0.020)}$ | $0.602_{(0.035)}$ | $0.373_{(0.043)}$ | $0.657_{(0.019)}$ |
| Gem | $0.584_{(0.041)}$ | $0.174_{(0.011)}$ | $0.201_{(0.065)}$ | $0.253_{(0.025)}$ | $0.367_{(0.025)}$ | $1.131_{(0.059)}$ | $0.290_{(0.019)}$ | $0.641_{(0.031)}$ | $0.551_{(0.023)}$ | $0.675_{(0.027)}$ |
| CIGIN | $0.564_{(0.057)}$ | $0.164_{(0.016)}$ | $0.254_{(0.010)}$ | $0.241_{(0.023)}$ | $0.347_{(0.023)}$ | $0.910_{(0.015)}$ | $0.318_{(0.020)}$ | $0.404_{(0.007)}$ | $0.411_{(0.032)}$ | $0.644_{(0.012)}$ |
| CGIB | $0.531_{(0.034)}$ | $0.156_{(0.014)}$ | $0.195_{(0.005)}$ | $0.203_{(0.033)}$ | $0.321_{(0.017)}$ | $0.892_{(0.022)}$ | $0.278_{(0.018)}$ | $0.391_{(0.006)}$ | $0.351_{(0.031)}$ | $0.613_{(0.023)}$ |
| I2Mole | $0.535_{(0.027)}$ | $0.158_{(0.011)}$ | $0.189_{(0.010)}$ | $0.175_{(0.014)}$ | $0.282_{(0.0011)}$ | $0.900_{(0.023)}$ | $0.268_{(0.013)}$ | $0.389_{(0.007)}$ | $0.306_{(0.030)}$ | $0.609_{(0.023)}$ |

Table 16: Replacing the current MPNN layer with standard backbones.

| GNN backbone | ACC (%) | Train time (s) | Test time (s) | Parameters (M) |
|---|---|---|---|---|
| GraphSAGE | 80.26 | 514.80 | 50.88 | 21.3 |
| GIN | 91.35 | 188.10 | 19.27 | 21.4 |
| GAT | 87.42 | 376.40 | 36.12 | 22.4 |
| Baseline | 95.34 | 1020.62 | 90.61 | 35.4 |

- An equally important aspect is that drugs often function only under specific conditions such as temperature and pH levels. Therefore, we anticipate future work to comprehensively consider the impact of external environments on the functionality of drug molecule pairs, thereby refining the model's capabilities.

## L  THE SCAFFOLD AND SIZE SPLITTING EXPERIMENTS RESULT.

The scaffold and size splitting experiments result are presented in Table 20.

In the scaffold split, we follow the standard practice used in molecular OOD evaluation Chung et al. (2022b); Yang et al. (2022b). We first match all molecules against a fixed set of SMARTS-defined scaffolds, consisting of nine predefined core substructure patterns in Table 19. All molecules containing any of these scaffolds are assigned to the test set. The remaining molecules are then randomly divided into training and validation sets using a 9:1 ratio. This ensures that structurally distinct scaffolds appear exclusively in the test domain.

In the size split, following the procedure in Ji et al. (2023), we group molecules according to their atomic size (number of atoms). Molecules are sorted in descending order of atomic size, and the ordered sequence is partitioned as follows: the largest 60% are assigned to the training set, the middle 20% to the validation set, and the smallest 20% to the test set.

## M  VISUALIZATION ANALYSIS

Acetaminophen, a widely used medication for pain relief (analgesic) and fever reduction (antipyretic), is frequently found in over-the-counter formulations. However, unexpected drug-drug interactions (DDIs) between acetaminophen and compounds such as Fenoterol, Fosphenytoin, and Ethanol can pose significant threats to patient safety, as depicted in Figure 7 (a). Specifically, the aromatic ring of acetaminophen, along with its surrounding functional groups, is capable of interacting with target molecules, particularly at the central carbon atom bonded to the carboxyl group. This key interaction has been effectively captured by the I2Mole model. Additionally, the core subgraphs extracted by I2Mole exhibit good connectivity and consistent distribution across regions, although the associated weights may vary.

As demonstrated in Figure 7 (b), In the case of aspirin, a widely used anti-inflammatory drug that also serves as an analgesic, antipyretic, and, at low doses, an antiplatelet agent, its interaction with

Table 17: Comparison of ChchMiner across Hypergraph Neural Network (HGNN) and I2mole. AM: average number of atoms per molecule.

| Model | Params (M) | AM=340 | AM=549 | AM=638 | AM=722 | AM=1934 | ACC(%) |
|---|---|---|---|---|---|---|---|
| I2Mole | 35.40 | 780.74 | 871.57 | 962.28 | 1012.22 | 1274.38 | 95.34 |
| HGNN | 13.225 | 350.27 | 437.63 | 456.94 | 478.81 | 499.75 | 92.37 |

Table 18: Comparison of ChchMiner across different models.

| GIN layer | ACC (%) | Decrease | Train time (s) | Test time (s) | Params (M) |
|---|---|---|---|---|---|
| 1 layer | 95.07 | 0.27 | 757.79 | 70.38 | 31.40 |
| 2 layer | 94.98 | 0.36 | 450.10 | 47.91 | 26.50 |
| 3 layer | 91.35 | 3.99 | 188.10 | 19.27 | 21.40 |
| 2layer+HGNN | 92.19 | 3.15 | 278.81 | 43.74 | 9.874 |
| **Baseline** | **95.34** | – | **1020.62** | **90.61** | **35.40** |

Table 19: SMARTS-defined scaffold patterns used in the scaffold split.

| | | |
|---|---|---|
| c1ccccc1C(=O)[O&H1] | C#[C&H1] | C[C&H2]Cl |
| C1=CCCC1 | c1ccc2c(-,:c1)ccc1ccccc12 | C1COCCN1 |
| NS(=O)(=O)C | c1ccccc1[N&+](=O)[O&-] | O[N+]([O-])=O |

molecules like Ibrutinib, Eluxadoline, and Glipizide is notably influenced by specific structural features. The aromatic branch of aspirin (excluding the carboxyl group region) is more prone to forming interactions with the nitrogen-containing heterocycles of other molecules. This suggests that, during DDI events, the merged graph substructures of these molecules have an enhanced propensity for direct interaction, leading to increased DDI potential. These observations provide important insights into the structural determinants of DDIs, further emphasizing the predictive capability of the I2Mole model in capturing complex inter-molecular relationships.

To further validate the interpretability of our model, we designed an evaluation strategy inspired by Zhong et al. (2024) to benchmark the alignment between model-identified substructures and experimentally supported chemical knowledge. Specifically, we curated a dataset of 73 chemicals (perpetrators) known to inhibit metabolic enzymes through well-defined functional groups, thereby inducing metabolism-mediated DDIs. These chemicals were paired with other drugs to generate 13,786 DDI instances. We evaluate the model at three complementary levels. **DDI-level** matching examines the classification performance across all 343,036 MMDDI pairs in the dataset, comprising 171,518 true interactions—where drug A inhibits or induces the metabolism of drug B—and an equal number of reverse-order negative samples generated by flipping the semantic roles of the two drugs. The model must determine whether the predicted mechanism and direction for each pair are correct, i.e., whether drug A truly affects drug B. **Perpetrator-level** matching further tests whether the model can correctly identify which drug in the pair acts as the perpetrator and which serves as the victim. Finally, **Frequent Functional Groups Matching** assesses substructure-level interpretability using a manually curated set of 73 chemicals known from the literature to cause metabolism-mediated DDIs through specific functional groups or reactive substructures.

Using I2Mole, we conducted interpretability analysis by comparing the substructures highlighted by our model with enzyme-inhibition functional groups reported in the literature. The results demonstrate that our model achieves a DDI-level matching rate of 60.63%, a perpetrator-level matching rate of 90.35%, and a frequent functional group matching rate of 78.42%, indicating that I2Mole captures key pharmacological substructures consistent with experimentally validated mechanisms.

Table 20: The scaffold and size splitting experiments result.

| Model | ZhangDDI | | Chchminer | | DeepDDI | |
|---|---|---|---|---|---|---|
| | Scaffold (↑) | Size (↑) | Scaffold (↑) | Size (↑) | Scaffold (↑) | Size (↑) |
| SSI-DDI | $72.34_{(3.73)}$ | $70.15_{(2.47)}$ | $72.34_{(1.22)}$ | $76.89_{(1.82)}$ | $78.27_{(2.12)}$ | $84.75_{(4.32)}$ |
| MDF-SA-DDI | $79.34_{(1.37)}$ | $79.46_{(0.48)}$ | $85.47_{(0.75)}$ | $84.23_{(0.63)}$ | $87.38_{(0.32)}$ | $86.58_{(0.37)}$ |
| DSN-DDI | $82.16_{(1.21)}$ | $80.38_{(1.23)}$ | $87.47_{(2.14)}$ | $87.92_{(1.32)}$ | $88.99_{(2.33)}$ | $86.53_{(1.65)}$ |
| CGIB | $83.32_{(1.26)}$ | $80.79_{(0.83)}$ | $89.47_{(1.10)}$ | $88.43_{(1.39)}$ | $89.56_{(2.22)}$ | $89.44_{(2.02)}$ |
| CMRL | $82.25_{(0.85)}$ | $81.32_{(0.77)}$ | $89.78_{(1.24)}$ | $88.49_{(1.91)}$ | $90.76_{(2.31)}$ | $90.77_{(1.22)}$ |
| IE-HGNN | $83.05_{(0.88)}$ | $81.67_{(0.69)}$ | $89.96_{(1.12)}$ | $89.02_{(1.37)}$ | $91.02_{(1.84)}$ | $91.05_{(1.14)}$ |
| IGIB-ISE | $83.22_{(0.79)}$ | $82.01_{(0.73)}$ | $90.11_{(1.05)}$ | $89.37_{(1.21)}$ | $91.25_{(1.77)}$ | $91.32_{(1.09)}$ |
| Ours | $\mathbf{83.45}_{(0.92)}$ | $\mathbf{82.55}_{(1.12)}$ | $\mathbf{90.32}_{(1.71)}$ | $\mathbf{89.95}_{(1.06)}$ | $\mathbf{91.76}_{(2.63)}$ | $\mathbf{91.89}_{(1.42)}$ |

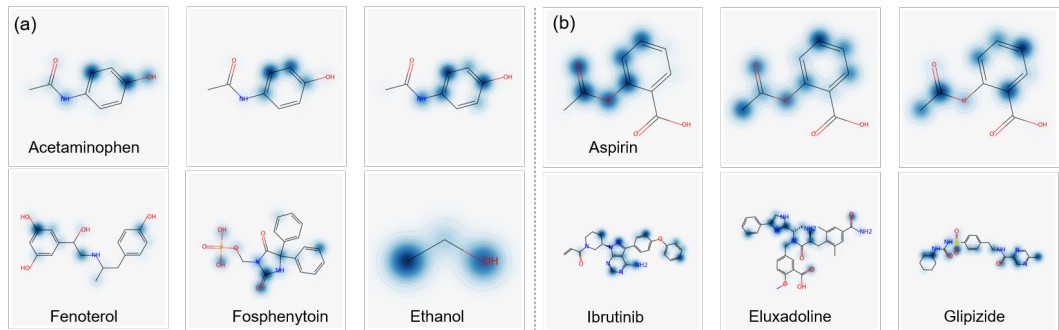

Figure 7: Visualization of the important substructure pairs in six drug pairs. (a) Acetaminophen with Fenoterol, Fosphenytoin, and Ethanol drug ligands. And (b) Aspirin with Ibrutinib, Eluxadoline, and Glipizide drug ligands. The darker the color means the greater the weight.

# N   MAXIMUM COMMON SUBSTRUCTURE (MCS) ANALYSIS OF VQ ENVIRONMENTS

We further assess whether the learned VQ codebook captures meaningful structural patterns by performing an MCS exemplar analysis. In particular, we examine environment categories 5, 6, and 7, which contain 4,556, 20,278, and 7,666 molecular pairs, respectively.

To evaluate the structural coherence of each VQ environment, we computed both inter-category and intra-category MCS similarities. Since MCS is computed at the molecular level and the overall number of molecules is large in our setting, the exact computation is computationally expensive. Therefore, we adopted a sampling-based evaluation strategy:

- **Inter-category analysis:** For each pair of categories, we randomly sampled 200 SMILES from each category. This yields 40,000 (200×200) cross-category molecular pairs. From these, we randomly selected 1,000 pairs and computed their MCS similarity. The resulting distribution reflects the structural overlap between the two categories.

- **Intra-category analysis:** For each of the three categories, we randomly sampled 200 SMILES from all molecules assigned to that category. From these 200 molecules, we randomly selected 500 molecular pairs and computed their MCS similarity. These 500 values characterize the structural consistency within the category.

Table 21: Interpretability evaluation results.

| Evaluation Aspect | Hit-Rate (%) |
|---|---|
| DDI-level Matching | 60.63 |
| Perpetrator-level Matching | 90.35 |
| Frequent Functional Groups Matching | 78.42 |

The MCS similarity between two molecules $G_1$ and $G_2$ is defined as:

$$S_{\text{MCS}} = \frac{|MCS|}{\min\left(|G_1|, |G_2|\right)}. \tag{32}$$

For each category, we report the mean and variance of the intra-category and inter-category MCS similarities, and for each pair of categories. The results are in Table 22.

Table 22: Inter-category and intra-category MCS similarity (mean ± variance).

| **Inter-Category** | Category 5 vs 6 | Category 5 vs 7 | Category 6 vs 7 |
|---|---|---|---|
| MCS similarity | $0.2523_{(0.1224)}$ | $0.2366_{(0.11014)}$ | $0.2491_{(0.1144)}$ |
| **Intra-Category** | Category 5 | Category 6 | Category 7 |
| MCS similarity | $0.3458_{(0.1259)}$ | $0.3921_{(0.1172)}$ | $0.3423_{(0.1109)}$ |

Our MCS-based analysis shows that the inter-category and intra-category structural similarities differ substantially. This indicates that the VQ-based clustering is not merely a direct grouping of molecules by shared substructures. Instead, the VQ codebook is learned in a latent embedding space, initialized from distinct environment vectors $env$, and each environment substructure vector $\widetilde{s}_{env}$ is mapped through a non-linear projection layer into the discrete code space as Equation 20. As a result, although the learned codewords reflect meaningful structural patterns, they are not expected to correspond one-to-one to MCS-defined structural clusters. As $\mathcal{L}_{vq}$ converges, the model obtains a stable codebook $\mathcal{W}$, which clusters the infinite possible environment space $\mathbf{E}$ into a discretized set of $M$ finite environments represented by $W$.

Furthermore, we extracted a representative MCS structure for each category. Specifically, for each category we randomly sampled 300 candidate molecules and computed the full 300×300 Tanimoto fingerprint similarity matrix. We then identified the "central" molecule, i.e., the one with the highest average Tanimoto similarity within the category and selected its top-40 nearest neighbors. Using RDKit, we computed the MCES shared by these 40 molecules. The resulting representative MCS patterns exhibit clear qualitative differences across categories: Category 5 tends to capture exocyclic C–C motifs, Category 6 is enriched in aromatic ring structures, and Category 7 is dominated by non-ring nitrogen atoms. These differences further confirm that the learned VQ codebook organizes molecular environments into semantically coherent and structurally distinct groups.

Table 23: Representative MCS exemplars extracted for each VQ environment category.

| Category | MCS exemplar (SMARTS) | Illusion |
|---|---|---|
| 5 | `[#6&R]-&!@[#6&!R]` | Exocyclic C–C |
| 6 | `6-member aromatic ring` | Aromatic rings |
| 7 | `[#7&!R]` | Non-ring nitrogen atom |

Overall, the VQ clustering results and the MCS analysis are not expected to align perfectly. Although the intra-category MCS values within each VQ cluster are relatively small, indicating that molecules in the same VQ category do not necessarily share large explicit common substructures, it is interesting to observe that each category nevertheless exhibits a distinct representative MCS pattern, and these patterns differ clearly across categories. This behavior is consistent with the fundamental difference between the two approaches: VQ clusters substructures based on learned semantic similarity in the latent embedding space, whereas MCS groups molecules purely according to graph-theoretic structural overlap. As a result, VQ could capture higher-level or functional similarities that may not correspond to large MCS fragments.

