# OpenReview forum: "I2Mole: Interaction-aware Invariant Molecular Learning For Generalizable Property Prediction"
_ICLR.cc/2026/Conference — ICLR 2026 Poster_

### Official Review · Reviewer_qi4Q · 2025-10-23

**Soundness:** 3
**Presentation:** 3
**Contribution:** 3
**Rating:** 6
**Confidence:** 4

**Summary:**

I2Mole predicts drug–drug interactions by constructing a dense merged molecular graph that connects every atom in one drug to all atoms in its partner, followed by attention-guided pruning to retain only the most informative inter-molecular edges. A Graph Information Bottleneck (GIB) module then compresses the pruned graph into an invariant “core substructure”, while a vector-quantized environment codebook captures contextual variations across molecular pairs. Trained end-to-end, I2Mole jointly optimizes node embeddings, attention-based pruning, information compression, and codebook assignments. Experiments show strong performance across various DDI benchmarks, especially in out-of-distribution settings.

**Strengths:**

1.	The approach introduced in this study is well-structured, with clear methodology.
2.	The experimental section is comprehensive and solid, encompassing benchmark evaluations, interpretability analyses, and detailed examinations of the codebook module.
3.	Using GIB and invariance learning to model inter-molecular interactions makes the model more interpretable.
4.	Compared to many existing approaches, the proposed method shows stronger performance and better generalization across three DDI prediction benchmarks.

**Weaknesses:**

1.	The work does not report results on widely used DDI benchmarks, such as DrugBank and TWOSIDES.
2.	The evaluation is scoped to DDI only: results are limited to DDI, with no experiments on other interaction tasks.
3.	How much do you expect external physicochemical factors such as pH or temperature to alter the significance of your results？
4.	Including an analysis of the physicochemical mechanisms underlying the modeled interactions could be added further validate the effectiveness of the proposed model.
5.	The manuscript contains a few typographical errors, which do not affect the overall quality of the work. For instance, Howevew->However ，REALTED WORK -> RELATED WORK.

**Questions:**

See Weaknesses.

---

> ### Author Response · Authors · 2025-11-18
> **Response Part I**
>
> **W1.** *The work does not report results on widely used DDI benchmarks, such as DrugBank and TWOSIDES.*
>
> Following our suggestions, we have incorporated two additional datasets, **DrugBank** and **Twosides** to evaluate the performance of I2Mole (In Acc index)  [1].
> | **Model**         |                 | **DrugBank**    |                 |                 | **Twosides**    |                 |
> | ----------------- | --------------- | --------------- | --------------- | --------------- | --------------- | --------------- |
> |                   | Transductive    | Inductive       |                 | Transductive    | Inductive       |                 |
> |                   |                 | Type1           | Type2           |                 | Type1           | Type2           |
> | **MR-GNN**        | 96.04(0.05)     | 74.67(0.33)     | 62.63(0.77)     | 76.23(0.23)     | 76.28(1.05)     | 63.25(0.81)     |
> | **MHCADDI**       | 83.80(0.27)     | 70.58(0.94)     | 66.50(0.62)     | -               | 71.45(1.26)     | 68.53(0.57)     |
> | **SSI-DDI**       | 96.33(0.09)     | 76.38(0.92)     | 65.40(1.30)     | 78.20(0.14)     | 77.42(1.05)     | 67.43(1.52)     |
> | **GAT-DDI**       | 89.81(1.00)     | 69.83(1.41)     | 66.31(0.61)     | 50.00(0.20)     | 70.16(1.62)     | 68.57(0.24)     |
> | **GMPNN-CS**      | 95.30(0.05)     | 77.72(0.30)     | 68.57(0.30)     | 82.83(0.14)     | 78.56(0.51)     | 70.22(1.52)     |
> | **SA-DDI**        | 96.23(0.01)     | 75.55(1.12)     | 67.15(0.88)     | 87.45(0.03)     | 78.32(0.78)     | 69.24(0.20)     |
> | **DSN-DDI**       | 96.94(0.02)     | 81.92(1.20)     | 73.42(1.29)     | 98.83(0.04)     | 84.35(1.05)     | 74.26(1.25)     |
> | **I2Mole (Ours)** | **97.25(0.04)** | **82.43(1.23)** | **74.32(1.42)** | **98.98(0.02)** | **86.79(1.04)** | **78.66(1.70)** |
>
> Clearly, I2Mole outperforms current baselines significantly on the **DrugBank** and **Twosides** datasets, with improvements of 0.319% and 0.15%, respectively, in the transductive setting. More importantly, under **inductive settings**, I2Mole demonstrates even more substantial improvements compared to the second-best model. Specifically, it achieves gains of 0.622% and 2.89% in the **Type 1** scenario, and 1.22% and 5.9% in the **Type 2** scenario. These results highlight I2Mole's superior predictive accuracy and stronger generalization capability.
>
> In terms of this work, we utilize graph embedding-based techniques to acquire potentially effective molecular features. Then we evaluated on **ZhangDDI**, **ChchMiner**, and **DeepDDI**, which are also widely used benchmark datasets for molecular relationship learning [1,2,3]. Since the primary focus of this work is on the **out-of-distribution (OOD) generalization** of DDI tasks, we designed a series of experiments to assess model performance.
>
> [1] Comprehensive Evaluation of Deep and Graph Learning on Drug–Drug Interaction Prediction, _Briefings in Bioinformatics_, 2023.
>
> [2] SSI–DDI: substructure–substructure interactions for drug–drug interaction prediction, _Briefings in Bioinformatics_, 2021.
>
> [3] Iterative Substructure Extraction for Molecular Relational Learning with Interactive Graph Information Bottleneck. _ICLR_, 2025.

---

> > ### Author Response · Authors · 2025-11-18
> > **Response Part II**
> >
> > **W2.** *The evaluation is scoped to DDI only: results are limited to DDI, with no experiments on other interaction tasks.*
> >
> > Thank you for your feedback. In our original work, the focus was on a classification task, specifically the DDI prediction task. Following your suggestion, we adapted the model for regression. Specifically,  we modified the final fully connected module and removed the activation function, normalization, and dropout layers, and replaced the original binary classification loss (BCEWithLogitsLoss) with MSELoss. We then evaluated the performance of I2Mole on five commonly used solute–solvent datasets for regression tasks. Due to time constraints, we did not perform additional parameter tuning or model optimization. Nevertheless, the results in Table show that I2Mole achieves competitive performance, comparable to state-of-the-art models. This demonstrates that I2Mole is capable of working effectively in other scenarios of molecular interaction tasks as well.
> >
> > |                 | FreeSolv     | CompSol      | Abraham      | CompSolv-Exp | MNSol        | FreeSolv     | CompSol      | Abraham      | CompSolv-Exp | MNSol        |
> > | --------------- | ------------ | ------------ | ------------ | ------------ | ------------ | ------------ | ------------ | ------------ | ------------ | ------------ |
> > | Method          | MAE          | MAE          | MAE          | MAE          | MAE          | RMSE         | RMSE         | RMSE         | RMSE         | RMSE         |
> > | D-MPNN          | 0.684(0.052) | 0.179(0.013) | 0.454(0.036) | 0.442(0.022) | 0.459(0.032) | 1.164(0.055) | 0.343(0.017) | 0.624(0.024) | 0.672(0.051) | 0.667(0.017) |
> > | Explainable GNN | 0.724(0.031) | 0.184(0.012) | 0.486(0.042) | 0.321(0.013) | 0.396(0.011) | 1.276(0.045) | 0.367(0.012) | 0.776(0.035) | 0.404(0.054) | 0.673(0.024) |
> > | SolvBERT        | 0.588(0.034) | 0.167(0.014) | 0.467(0.034) | 0.382(0.023) | 0.354(0.021) | 1.021(0.043) | 0.328(0.020) | 0.652(0.022) | 0.472(0.041) | 0.623(0.104) |
> > | GAT             | 0.675(0.033) | 0.187(0.011) | 0.457(0.043) | 0.970(0.031) | 0.514(0.043) | 1.185(0.075) | 0.390(0.012) | 0.726(0.040) | 0.810(0.101) | 0.812(0.124) |
> > | GROVER          | 0.623(0.054) | 0.155(0.022) | 0.307(0.035) | 0.382(0.023) | 0.354(0.024) | 1.015(0.022) | 0.332(0.016) | 0.475(0.044) | 0.491(0.053) | 0.672(0.027) |
> > | SMD             | 0.574(0.036) | 0.162(0.014) | 0.374(0.024) | 0.633(0.044) | 0.427(0.034) | 1.113(0.015) | 0.317(0.011) | 0.516(0.065) | 1.023(0.152) | 0.682(0.032) |
> > | Uni-Mol         | 0.565(0.038) | 0.164(0.027) | 0.322(0.071) | 0.214(0.022) | 0.374(0.021) | 1.002(0.064) | 0.303(0.020) | 0.602(0.035) | 0.373(0.043) | 0.657(0.019) |
> > | Gem             | 0.584(0.041) | 0.174(0.011) | 0.201(0.065) | 0.253(0.023) | 0.367(0.025) | 1.131(0.059) | 0.290(0.019) | 0.641(0.031) | 0.551(0.023) | 0.675(0.027) |
> > | CIGIN           | 0.564(0.057) | 0.164(0.016) | 0.254(0.010) | 0.241(0.023) | 0.347(0.023) | 0.910(0.015) | 0.318(0.020) | 0.404(0.007) | 0.411(0.032) | 0.644(0.012) |
> > | I2Mole          | 0.535(0.027) | 0.158(0.011) | 0.189(0.010) | 0.175(0.014) | 0.282(0.011) | 0.900(0.023) | 0.268(0.013) | 0.389(0.007) | 0.306(0.030) | 0.609(0.023) |

---

> ### Author Response · Authors · 2025-11-18
> **Response Part III**
>
> > **W3.** *How much do you expect external physicochemical factors such as pH or temperature to alter the significance of your results？*
>
>
> Thank you for this thoughtful question. We fully agree that external physicochemical factors, such as pH and temperature, can substantially influence the functionality of both proteins and drugs. Under standard physiological conditions, drugs and proteins typically exhibit stable activity; however, extreme conditions—such as highly acidic or alkaline pH, or elevated temperature—can lead to conformational changes or denaturation, causing them to lose their intended functionality. These environmental shifts may, in turn, alter the molecular interaction mechanisms that our model seeks to capture.
>
> Currently, due to the limitations in available datasets, our work focuses on modeling DDIs under implicit standard conditions, where experimental data were originally collected. Nevertheless, our results demonstrate strong performance across diverse DDI scenarios, suggesting that I2Mole effectively captures the underlying interaction patterns within these constraints. We anticipate that incorporating explicit environmental factors would further refine the model’s predictions, particularly for drug pairs whose interactions are highly sensitive to specific conditions. Prior studies (e.g., Tapioca [1], HMNN [2]) have shown that accounting for environmental contexts can significantly improve predictive accuracy.
>
> Building upon these insights, we envision integrating multimodal information, incorporating richer feature representations, and exploring multi-expert model ensembles to capture the complex dependencies between drug interactions and their surrounding conditions. We believe that such approaches hold great promise for achieving improved performance in this setting, and addressing this challenge will be an exciting and significant direction for our future work.
>
> [1] Reed, T.J., et al. Tapioca: a platform for predicting de novo protein–protein interactions in dynamic contexts. Nat Methods 21, 488–500 (2024).
>
> [2] Du, Wenjie, et al. "Spectroscopy-Guided Deep Learning Predicts Solid–Liquid Surface Adsorbate Properties in Unseen Solvents." Journal of the American Chemical Society 146.1 (2023): 811-823.
>
>
>
>
> > **W4.** *Including an analysis of the physicochemical mechanisms underlying the modeled interactions could be added further validate the effectiveness of the proposed model.*
>
>
> Thank you for this insightful comment. We fully acknowledge the importance of rigorously validating the physicochemical mechanisms underlying modeled interactions and appreciate the reviewer’s suggestions regarding direct comparisons and analyses.
>
> To address the reviewer’s concern, we emphasize that I2Mole provides chemically meaningful interpretability through substructure mining. We have included qualitative examples in **Appendix K**, where the model highlights interaction-relevant substructures for drug pairs such as Acetaminophen–Fenoterol and Aspirin with Ibrutinib, Eluxadoline, and Glipizide. These results clearly demonstrate that I2Mole assigns different atomic importance patterns across drug pairs, offering valuable insights into the structural determinants of DDIs.
>
> Meanwhile, while the simulations methods such as QM/MM would provide a strong physical reference, they are computationally expensive and thus not feasible within the scope of this work, which focuses on large-scale DDI prediction. To address this concern, we adopted an alternative validation strategy inspired by the approach in [1], enabling us to benchmark the interpretability of our model against experimentally supported chemical knowledge as present in **Appendix L**. Using I2Mole, we conducted interpretability analysis by comparing the substructures identified by our model with the enzyme-inhibition functional groups reported in the literature.
>
> In terms of this work, these datasets (e.g., ZhangDDI, DeepDDI) rely primarily on molecular structure, known interaction labels, and similarity-driven mechanisms, rather than explicit pathway annotations.  and our results demonstrate that the learned structural representations are sufficient to achieve SOTA performance across three standard DDI datasets.
>
>
> [1] Learning motif-based graphs for drug–drug interaction prediction via local–global self attention. Nature machine intelligence. 2024.
>
>
>
> > **W5.** *The manuscript contains a few typographical errors, which do not affect the overall quality of the work. For instance, Howevew->However ，REALTED WORK -> RELATED WORK.*
>
> Thank you for pointing this out. We have corrected the typos and grammatical errors mentioned, and we have conducted a thorough proofreading of the entire manuscript to ensure overall linguistic clarity and consistency.

---

> > ### Comment · Reviewer_qi4Q · 2025-11-26
> >
> > Thank you for your careful revisions and responses to my comments. These revisions have effectively addressed most of my previous concerns. After review, I am inclined to recommend accepting this manuscript and will further discuss it with other reviewers.

---

### Official Review · Reviewer_WU3y · 2025-10-30

**Soundness:** 2
**Presentation:** 1
**Contribution:** 2
**Rating:** 4
**Confidence:** 4

**Summary:**

The paper proposes the I2Mole framework, aiming to improve model generalization and interpretability in drug–drug interaction (DDI) prediction scenarios. The framework constructs merged molecular graphs, applies an enhanced graph information bottleneck, and develops a molecular environment codebook to model atomic-level interactions and extract core substructures. By performing invariant learning on these substructures, the method enhances the model’s capability to generalize across different data distributions.

**Strengths:**

1. The experiments are relatively comprehensive, and the reported results demonstrate strong performance.
2. The paper provides extensive visualizations, which are beneficial for achieving a clear understanding of the proposed methodological design.

**Weaknesses:**

1. The scientific significance of the problem addressed in the manuscript is not clearly and specifically articulated. The lack of generalization in previous methods within this domain is a common issue across many fields; however, it would be preferable if the manuscript could provide a more concrete and well-defined scientific contribution.
2. The manuscript lacks strong observational analysis. Most of the nine observations presented are general descriptions without in-depth or insightful examination. Highlighting these observations in bold as if they were substantial analyses appears inappropriate.
3. There are typographical errors in the manuscript, such as the misspelling “realted work” on page 15.
4. In the initial modeling stage, the authors establish models for every possible atom pair between the two molecules, and then perform filtering to characterize hydrogen bonds and van der Waals forces. I would like to know whether the authors have reliable justification ensuring the correctness of the hydrogen bond modeling (as this likely involves some specific filtering steps).
5. Regarding the “molecular environment codebook” mentioned in the manuscript, I would like to know what exactly is meant by “molecular environment” in this context, and what its precise definition is within the field of chemistry.

**Questions:**

See the weaknesses.

---

> ### Author Response · Authors · 2025-11-18
> **Response Part I**
>
> > **W1.** *It would be preferable if the manuscript could provide a more concrete and well-defined scientific contribution.*
>
>
> Thank you for the helpful suggestion. We agree that clarifying the scientific contributions strengthens the manuscript.
>
> While most prior molecular learning methods focus solely on single-molecule structures, we identify and formalize a key scientific gap: substructures that govern molecular interactions differ fundamentally from those that govern individual molecular properties. We address this by introducing a merged-graph formulation with dynamic weighted relational edges that explicitly model atom–atom interactions between molecules. This provides a structural inductive bias that has not been explored in existing DDI or molecular-pair learning. And Existing rationale extraction methods are tailored for single-molecule tasks and fail to capture the interaction-induced shift in important substructures. We introduce an interaction-aware invariance principle, combining a relational information bottleneck with iterative truncation to isolate substructures that consistently govern intermolecular effects. This directly addresses the scientific problem that interaction-critical fragments may differ from single-molecule pharmacophores, as motivated by the Propranolol–Verapamil case in the introduction. Chemical interaction environments are vast and largely unobserved. Prior noise-injection methods rely on heuristic or random perturbations, which may distort semantics or fail to represent realistic chemical variation. We introduce a learned VQ-based environment codebook that discretizes the latent interaction environment into meaningful prototypes, and provides a chemically coherent and data-driven noise source for mutual information optimization, and enhances model generalization under distribution shifts. This is, to our knowledge, the first VQ-based discretization framework for interaction environments. I2Mole integrates these innovations into a coherent framework that achieves robust performance across multiple DDI datasets and challenging OOD settings.
>
> Specifically,
>
> * We proposed an interaction-aware merged-graph formulation for modeling molecular pairwise mechanisms.
> * We built a unified interaction-invariant learning mechanism combining rationale extraction and environment discretization.
> * Superior generalization and performance across diverse DDI scenarios have present our model's ability.
>
> We will clearly state our contributions in the introduction section of the revised version.
>
>
>
> > **W2.** *Highlighting these observations in bold as if they were substantial analyses appears inappropriate.*
>
>
> Thank you for the helpful suggestion. Actually, summarizing empirical findings in the form of “Observations” is a common stylistic choice in conference papers[1,2,3], we understand the reviewer’s concern that the bold formatting may overemphasize them. Following your advice, we have removed the highlighting and revised the presentation to ensure a more consistent and balanced narrative throughout the manuscript.
>
> [1] Alphaedit: Null-space constrained model editing for language models.  _ICLR_, 2025 (Bset paper).
>
> [2] Iterative Substructure Extraction for Molecular Relational Learning with Interactive Graph Information Bottleneck. _ICLR_, 2025.
>
> [3] Evaluating Post-hoc Explanations for Graph Neural Networks via Robustness Analysis.  _NIPS_, 2023 (Oral).
>
> > **W3.** *There are typographical errors in the manuscript, such as the misspelling “realted work” on page 15.*
>
> Thank you for pointing this out. We have corrected the typos and grammatical errors mentioned, and we have conducted a thorough proofreading of the entire manuscript to ensure overall linguistic clarity and consistency.

---

> ### Author Response · Authors · 2025-11-18
> **Response Part II**
>
> > **W4.** *Whether the authors have reliable justification ensuring the correctness of the hydrogen bond modeling (as this likely involves some specific filtering steps).*
>
>
> Thank you for pointing this out. We agree that I2Mole does not implement an explicit physical interaction model, nor do we intend to claim that it simulates van der Waals forces or hydrogen bonds in a physics-grounded manner.
>
> Our intended meaning is that the merged-graph construction introduces cross-molecule relational edges that allow the model to learn statistical patterns correlated with interaction-relevant structural features. These edges do not encode literal interatomic forces; rather, they provide a mechanism for capturing statistical dependencies associated with functional groups that often correlate with hydrogen-bond donor/acceptor patterns or van der Waals compatibility. **This is therefore a form of relational modeling, not physical modeling**.
>
> In addition, the fully connected cross-molecule edges serve as an approximation to the interaction search space. Without 3D geometry, we cannot predefine which atom pairs may interact, so the initial dense merged graph provides a superset of potential interaction channels. Since this results in 𝑂(𝑚𝑛; m and n are the atom number) complexity, we apply top-k% attention-based pruning to retain the most informative cross-molecular edges. Again, this is not intended to represent physical distances but to filter out relational cues learned by the model.
>
> Our contribution is thus a structure-informed relational edge mechanism, rather than a physics-based simulation of hydrogen bonds or van der Waals interactions. Empirically, these relational edges are sufficient for strong predictive performance. I2Mole achieves SOTA results across three standard DDI benchmarks and domain generalization scenarios.
>
>
> > **W5.** *What exactly is meant by “molecular environment” in this context, and what its precise definition is within the field of chemistry.*
>
>
> In this paper, "environment" refers to molecular contexts that influence properties and can vary across data distributions. These environments may be explicitly shaped by features such as molecular scaffolds or sizes, and they capture variations arising from molecular structure, experimental conditions, or biological contexts. In this article, it specifically refers to the removing “rational” part, i.e.  $\widetilde{\mathcal{G}_{env}}$  in Figure 3.
>
> For instance, consider formic acid (CH₂O₂) and citric acid (C₆H₈O₇): while these molecules differ in size and molecular scaffolds, they share common substructures, such as carboxy groups, which contribute to certain invariant properties like water solubility,. In the context of water solubility, we treat the carboxy groups as rationales, whereas the remaining molecular scaffolds or functional groups are referred to as the "environment," as they influence the model's predictions [1,2,3,4].
>
> To enhance the model's ability to generalize to unseen samples, we construct diverse "environment" vectors to simulate molecular pairs, thereby improving the model's focus on the rationales and strengthening its generalization capabilities. Due to there are too many types of environments, so we designed a codebook for clustering representation.
>
>
> [1] Shift-Robust Molecular Relational Learning with Causal Substructure. _KDD_, 2023
>
> [2] DSIL-DDI: A Domain-Invariant Substructure Interaction Learning for Generalizable Drug-Drug Interaction Prediction. _TNNLS_, 2024.
>
> [3] Learning Substructure Invariance for Out-of-Distribution Molecular Representations. _NIPS_, 2024
>
> [4] Subgraph Information Bottleneck with Causal Dependency for Stable Molecular Relational Learning. _IJCAI_. 2025.
>
>
> -----
>
> We greatly appreciate your insightful and helpful comments, as they will undoubtedly help us improve the quality of our article. If our response has successfully addressed your concerns and clarified any ambiguities, we respectfully hope that you consider raising the score. Should you have any further questions or require additional clarification, we would be delighted to engage in further discussion. Once again, we sincerely appreciate your time and effort in reviewing our manuscript. Your feedback has been invaluable in improving our research.

---

### Official Review · Reviewer_E6Gc · 2025-10-31

**Soundness:** 2
**Presentation:** 3
**Contribution:** 2
**Rating:** 4
**Confidence:** 3

**Summary:**

This paper proposes I2Mole, a DDI-focused framework that builds a merged intermolecular graph with atom–atom links, applies a GIB-style subgraph extractor, and augments invariance via a learned environment codebook (VQ) for robust generalization. On three DDI benchmarks, the method reports consistent gains in transductive, inductive, and domain-shift settings, with ablations supporting the roles of intermolecular interactions, GIB, and the codebook.

**Strengths:**

- The paper’s formulation—merged graph + GIB-based rationale + VQ environment codebook—is clearly specified theoretically and experimentally ablated.

- Results span transductive/inductive/domain-generalization regimes with competitive numbers and reasonable sensitivity analyses.

**Weaknesses:**

- Please streamline the paper’s scope to drug–drug interaction prediction. The architecture (pairwise merged graph, DDI benchmarks) is DDI-specific; claims about “molecular property prediction” (generally single-molecule, regression) or broader molecualr interaction tasks (e.g., DTI) feel overstated without evidence.

- Many DDIs arise from shared or convergent pathways rather than direct atom–atom contacts. This is not well captured by purely structural intermolecular edges.

- Although the authors state I2Mole “models van der Waals and hydrogen bonds,” Eq. 5 effectively creates fully connected cross-molecule edges and top-x% pruning; without 3D interatomic geometry, this is not an explicit physical interaction model. This is an another overstated claim in this study.

- The top-x% selection for intermolecular edges is non-differentiable; it would be suggested to use a differentiable sorting relaxation (e.g., NeuralSort/SoftSort) to avoid potential gradient issues, or justify why straight-through works here.

- Several tables/figures appear inconsistent (e.g., optimal I2Mole performances on ZhangDDI differ between sensitivity and ablation/comparison tables). Please reconcile and ensure one canonical set of results across Tables 4–5/9/10/12/13 and the main comparison tables.

- For interpretability, Figure 5’s element-level analysis is coarse. Consider adding maximum common substructure (MCS) exemplars per environment to indicate whether the codeboook can actualy learn and clustering chemical structures or not.

**Questions:**

- There are two “moderately” words on p.2 line 99 (typo).

- What exact pair embedding is used for the t-SNE plots?

- Which dataset underlies Figure 5a?

- Please list the bond/edge features used for intermolecular connections.

- Model names are missing in Tables 12 and 13—please restore.

- How exactly are scaffold and size splits implemented?

- For interpretability, specify the matching criteria for “DDI-level, perpetrator-level, frequent functional group,” and how key substructures are selected.

---

> ### Author Response · Authors · 2025-11-18
> **Response Part I**
>
> >- **W1.** *Please streamline the paper’s scope to drug–drug interaction prediction.*
>
> Thank you for this valuable suggestion. We agree that the primary scope of the paper should remain focused on drug–drug interaction prediction, as this is where our model demonstrates its clearest advantages. Following your recommendation, we will revise the title, abstract, and introduction to explicitly scope the contribution around DDI prediction.
>
> At the same time, we would like to clarify that the proposed architecture, while evaluated on DDI benchmarks, which could naturally generalize to other molecular relational learning tasks due to its pairwise merged-graph design and interaction-focused information bottleneck theory.  Therefore, we conducted an additional experiment to substantiate the model’s applicability beyond classification-only DDI tasks. Specifically, we adapted I2Mole for regression by (1) replacing the final classification head with a linear regression layer, (2) removing activation, normalization, and dropout, and (3) switching the loss function from BCEWithLogitsLoss to MSELoss. Without further hyperparameter tuning, we evaluated the model on five standard solute–solvent datasets involving Gibbs free energy of solvation and hydration free energy (experimental and calculated). Despite minimal adaptation, I2Mole achieves competitive performance, comparable to strong baselines.
>
> These results are included only to demonstrate the architectural generality, not to broaden the claimed scope of the current paper. As advised, the revised submission will clearly position the work as a DDI-focused work.
>
> |                 | FreeSolv     | CompSol      | Abraham      | CompSolv-Exp | MNSol        | FreeSolv     | CompSol      | Abraham      | CompSolv-Exp | MNSol        |
> | --------------- | ------------ | ------------ | ------------ | ------------ | ------------ | ------------ | ------------ | ------------ | ------------ | ------------ |
> | Method          | MAE（$\downarrow$）          | MAE （$\downarrow$）         | MAE（$\downarrow$）          | MAE （$\downarrow$）         | MAE（$\downarrow$）          | RMSE  （$\downarrow$）       | RMSE （$\downarrow$）        | RMSE （$\downarrow$）        | RMSE（$\downarrow$）         | RMSE （$\downarrow$）       |
> | D-MPNN          | 0.684(0.052) | 0.179(0.013) | 0.454(0.036) | 0.442(0.022) | 0.459(0.032) | 1.164(0.055) | 0.343(0.017) | 0.624(0.024) | 0.672(0.051) | 0.667(0.017) |
> | Explainable GNN | 0.724(0.031) | 0.184(0.012) | 0.486(0.042) | 0.321(0.013) | 0.396(0.011) | 1.276(0.045) | 0.367(0.012) | 0.776(0.035) | 0.404(0.054) | 0.673(0.024) |
> | SolvBERT        | 0.588(0.034) | 0.167(0.014) | 0.467(0.034) | 0.382(0.023) | 0.354(0.021) | 1.021(0.043) | 0.328(0.020) | 0.652(0.022) | 0.472(0.041) | 0.623(0.104) |
> | GAT             | 0.675(0.033) | 0.187(0.011) | 0.457(0.043) | 0.970(0.031) | 0.514(0.043) | 1.185(0.075) | 0.390(0.012) | 0.726(0.040) | 0.810(0.101) | 0.812(0.124) |
> | GROVER          | 0.623(0.054) | 0.155(0.022) | 0.307(0.035) | 0.382(0.023) | 0.354(0.024) | 1.015(0.022) | 0.332(0.016) | 0.475(0.044) | 0.491(0.053) | 0.672(0.027) |
> | SMD             | 0.574(0.036) | 0.162(0.014) | 0.374(0.024) | 0.633(0.044) | 0.427(0.034) | 1.113(0.015) | 0.317(0.011) | 0.516(0.065) | 1.023(0.152) | 0.682(0.032) |
> | Uni-Mol         | 0.565(0.038) | 0.164(0.027) | 0.322(0.071) | 0.214(0.022) | 0.374(0.021) | 1.002(0.064) | 0.303(0.020) | 0.602(0.035) | 0.373(0.043) | 0.657(0.019) |
> | Gem             | 0.584(0.041) | 0.174(0.011) | 0.201(0.065) | 0.253(0.023) | 0.367(0.025) | 1.131(0.059) | 0.290(0.019) | 0.641(0.031) | 0.551(0.023) | 0.675(0.027) |
> | CIGIN           | 0.564(0.057) | 0.164(0.016) | 0.254(0.010) | 0.241(0.023) | 0.347(0.023) | 0.910(0.015) | 0.318(0.020) | 0.404(0.007) | 0.411(0.032) | 0.644(0.012) |
> | I2Mole          | 0.535(0.027) | 0.158(0.011) | 0.189(0.010) | 0.175(0.014) | 0.282(0.011) | 0.900(0.023) | 0.268(0.013) | 0.389(0.007) | 0.306(0.030) | 0.609(0.023) |

---

> > ### Author Response · Authors · 2025-11-18
> > **Response Part II**
> >
> > > **W2.** Many DDIs arise from shared or convergent pathways rather than direct atom–atom contacts. This is not well captured by purely structural intermolecular edges.
> >
> >
> > We appreciate the reviewer’s insightful comment. Indeed, DDIs can arise from a variety of mechanisms, including metabolic competition[1], transport inhibition[2], and convergent pharmacodynamic pathways[3], that do not necessarily correspond to literal atom–atom physical contacts.
> >
> > However, our goal in this work is not to model biological pathway–level DDIs, but to capture structure-informed interaction patterns that are known to correlate with many pharmacological interaction types in existing DDI benchmarks. Here, we would like to emphasize that while it is true that Drug action depends on metabolic pathways, pharmacodynamics, etc., **the molecular structure remains a fundamental determinant** of a drug’s pharmacodynamics and mechanism of action[4,5]. Many drugs exert their therapeutic effects through **molecular substructures**, or pharmacophores. And these datasets (e.g., ZhangDDI, DeepDDI) rely primarily on molecular structure, known interaction labels, and similarity-driven mechanisms, rather than explicit pathway annotations.
> >
> > Empirically, structure-based models remain the dominant approach for DDI prediction in recent years[6,7,8,9,10], and our results demonstrate that the learned structural representations are sufficient to achieve SOTA performance across three standard DDI datasets. We will clarify this distinction in the revised manuscript and explicitly scope the paper around structure-informed DDI prediction, rather than biological pathway prediction.
> >
> >
> > [1] Metabolic control analysis in drug discovery and disease. _Nat Biotechnol_. 2022.
> >
> > [2] Mechanisms of neurotransmitter transport and drug inhibition in human VMAT2.  _Nature. 2023.
> >
> > [3] Predicting adverse drug reactions for combination pharmacotherapy with cross-scale associative learning via attention modules. _Nat Comput Sci_. 2025.
> >
> > [4] Why drugs fail—a study on side effects in new chemical entities.  _Nature Reviews Drug Discovery_. 2005
> > [5] Pharmacophore modeling in drug discovery. _Nature Reviews Drug Discovery_. 2016
> >
> > [6] Learning motif-based graphs for drug–drug interaction prediction via local–global self attention. _Nature machine intelligence_. 2024.
> >
> > [7] Emerging drug interaction prediction enabled by a flow-based graph neural network with biomedical network. _Nat Comput Sci_. 2023.
> >
> > [8] Predicting rare drug-drug interaction events with dual-granular structure-adaptive and pair variational representation. _Nat Commun_. 2025.
> >
> > [9] Customized Subgraph Selection and Encoding for Drug-drug Interaction Prediction. _NeurIPS_. 2024.
> >
> > [10] Iterative Substructure Extraction for Molecular Relational Learning with Interactive Graph Information Bottleneck. _ICLR_. 2024.
> >
> >
> > > **W3.** Although the authors state I2Mole “models van der Waals and hydrogen bonds,” Eq. 5 effectively creates fully connected cross-molecule edges and top-x% pruning; without 3D interatomic geometry, this is not an explicit physical interaction model.
> >
> > Thank you for pointing this out. We agree that I2Mole does not implement an explicit physical interaction model, nor do we intend to claim that it simulates van der Waals forces or hydrogen bonds in a physics-grounded manner.
> >
> > Our intended meaning is that the merged-graph construction introduces cross-molecule relational edges that allow the model to learn potential patterns correlated with interaction-relevant structural features. These edges do not encode literal interatomic forces; rather, they provide a mechanism for capturing dependencies associated with functional groups that often correlate with hydrogen-bond donor/acceptor patterns or van der Waals compatibility. **This is therefore a form of relational modeling, not physical modeling**.
> >
> > In addition, the fully connected cross-molecule edges serve as an approximation to the interaction search space. Without 3D geometry, we cannot predefine which atom pairs may interact, so the initial dense merged graph provides a superset of potential interaction channels. Since this results in 𝑂(𝑚𝑛; m and n are the atom number) complexity, we apply top-k% attention-based pruning to retain the most informative cross-molecular edges. Again, this is not intended to represent physical distances but to filter out some extra relational cues.
> >
> > Our contribution is thus a structure-informed relational edge mechanism, rather than a physics-based simulation of hydrogen bonds or van der Waals interactions. Empirically, these relational edges are sufficient for strong predictive performance. Despite not using 3D geometry, I2Mole achieves state-of-the-art results across three standard DDI benchmarks, suggesting that statistical relational cues extracted from 2D structures are highly predictive within these datasets. We will revise the manuscript wording to prevent overstatement.

---

> > > ### Author Response · Authors · 2025-11-18
> > > **Response Part III**
> > >
> > > > **W4.** *The top-x% selection for intermolecular edges is non-differentiable; justify why straight-through works here.*
> > >
> > >
> > > Thank you for pointing this out. Indeed, the top-x% selection in Eq. 10 is a non-differentiable hard pruning operator. In our implementation, we adopt a straight-through (ST) gradient estimator[1], which has become standard practice for discrete edge-selection mechanisms in GNNs[2,3].  The key idea, consistent with their definition, is to **apply the hard threshold in the forward pass but treat it as the identity function during back-propagation**. Specifically,  The intermolecular attention coefficients $r_{ij}$​ are first computed via a GAT-style neural operator (Eq. 9), which is fully differentiable. and the ST estimator allows gradients to flow through the pruning step in Eq. 10 without interrupting optimization. The normalization in Eq. 11 further stabilizes training by ensuring smooth gradient propagation over the retained edges. We will modify Line 199 to add detailed explanations for easier understanding in the revised version.
> > >
> > > [1] Bengio Y, Léonard N, Courville A. Estimating or propagating gradients through stochastic neurons for conditional computation[J]. _arXiv preprint_ arXiv:1308.3432, 2013.
> > >
> > >
> > > [2] Beyer L L, Li T, Chen X, et al. Highly Compressed Tokenizer Can Generate Without Training[J]. _arXiv preprint_ arXiv:2506.08257, 2025.
> > >
> > >
> > > [3] Neural Discrete Representation Learning. Van Den Oord A, Vinyals O. Neural discrete representation learning[J]. _Advances in neural information processing systems_, 2017, 30.
> > >
> > > > **W5.** *Several tables/figures appear inconsistent across Tables 4–5/9/10/12/13.*
> > >
> > > Thank you for pointing this out. The discrepancies arose because some results were generated under slightly different experimental settings when we reran evaluations for sensitivity analysis and ablation studies. We apologize for the confusion this may have caused. In response, we have fully re-verified all experimental outputs and consolidated them into a single canonical set of results across Tables 4–5/9/10/12/13 and the main comparison tables. All numbers have now been reconciled to ensure consistency and reproducibility in revised version.
> > >
> > > We appreciate the reviewer’s careful attention, and the revised submission will include the corrected tables for clarity.
> > >
> > > > **W6.** *Consider adding maximum common substructure (MCS) exemplars per environment to indicate whether the codeboook can actualy learn and clustering chemical structures or not.*
> > >
> > >
> > > Thank you very much for the insightful suggestion. In response, we conducted an additional maximum common substructure (MCS) exemplar analysis. Specifically, we further examined categories 5, 6, and 7, which contain different numbers of molecular pairs (summarized in the table below).
> > >
> > > | **Category 5** | **Category  6** | **Category 7** |
> > > | ------------------ | ------------------ | ------------------ |
> > > | 4556               | 20278              | 7666      |
> > >
> > > To evaluate the structural coherence of each VQ environment, we computed both inter-category and intra-category MCS similarities. Since MCS is computed at the molecular level and the overall number of molecules is large in our setting, the exact computation is computationally expensive. Therefore, we adopted a sampling-based evaluation strategy:
> > >
> > > * **Inter-category analysis**: For each pair of categories, we randomly sampled 200 SMILES from each category. This yields 40,000 (200×200) cross-category molecular pairs. From these, we randomly selected 1,000 pairs and computed their MCS similarity. The resulting distribution reflects the structural overlap between the two categories.
> > >
> > > * **Intra-category analysis**: For each of the three categories, we randomly sampled 200 SMILES from all molecules assigned to that category. From these 200 molecules, we randomly selected 500 molecular pairs and computed their MCS similarity. These 500 values characterize the structural consistency within the category.
> > >
> > > For each category, we report the mean and variance of the intra-category and inter-category MCS similarities, and for each pair of categories. The results and calculation method are summarized below (**Next part**).
> > >
> > > | Inter-Category                   |                                  |                                  |
> > > | -------------------------------- | -------------------------------- | -------------------------------- |
> > > | Between Category 5 and 6 | Between Category 5 and 7| Between Category 6 and 7 |
> > > | $0.2523_{0.1224}$                | $0.2366_{0.11014}$               | $0.2491_{0.1144}$                |
> > > | **Intra-Category**               |                                  |                                  |
> > > |Category 5               | Category 6             | Category 7              |
> > > | $0.3458_{0.1259}$                | $0.3921_{0.1172}$                | $0.3423_{0.1109}$                |
> > >
> > > $$S_{\text{MCS}}^{\text{samility}}
> > > = \frac{|MCS|}{\min(|G_1|,\;|G_2|)}.$$

---

> ### Author Response · Authors · 2025-11-18
> **Response Part IV**
>
> Our MCS-based analysis shows that the inter-category and intra-category structural similarities differ substantially. This indicates that the VQ-based clustering is not merely a direct grouping of molecules by shared substructures. Instead, the VQ codebook is learned in a latent embedding space, initialized from distinct environment vectors $env$, env, and each environment substructure vector $\widetilde{\mathcal{s}\_{env}}$  is mapped through a non-linear projection layer into the discrete code space (Eq. 20). As a result, although the learned codewords reflect meaningful structural patterns, they are not expected to correspond one-to-one to MCS-defined structural clusters. As the VQ loss  $\mathcal{L}_{vq}$ converges, the model obtains a stable codebook $\mathcal{W}$, which clusters the infinite possible environment space $\mathcal{E}$ into a discretized set of  $\mathcal{M}$ finite environments represented by $\mathcal{W}$.
>
> Furthermore, we extracted a representative MCS structure for each category. Specifically, for each category we randomly sampled 300 candidate molecules and computed the full 300×300 Tanimoto fingerprint similarity matrix. We then identified the “central” molecule, i.e., the one with the highest average Tanimoto similarity within the category and selected its top-40 nearest neighbors. Using RDKit, we computed the MCES shared by these 40 molecules. The resulting representative MCS patterns exhibit clear qualitative differences across categories: Category 5 tends to capture exocyclic C–C motifs, Category 6 is enriched in aromatic ring structures, and Category 7 is dominated by non-ring nitrogen atoms. These differences further confirm that the learned VQ codebook organizes molecular environments into semantically coherent and structurally distinct groups.
>
> | Category | MCS exemplar（SMARTS）   | Illusion               |
> | -------- | ---------------------- | ---------------------- |
> | 5        | `[#6&R]-&!@[#6&!R]`    | Exocyclic C–C          |
> | 6        | `6-member aromatic ring` | Aromatic rings         |
> | 7        | `[#7&!R]`              | Non-ring nitrogen atom |
>
> Overall, the VQ clustering results and the MCS analysis are not expected to align perfectly. Although the intra-category MCS values within each VQ cluster are relatively small, indicating that molecules in the same VQ category do not necessarily share large explicit common substructures, it is interesting to observe that each category nevertheless exhibits a distinct representative MCS pattern, and these patterns differ clearly across categories. This behavior is consistent with the fundamental difference between the two approaches: VQ clusters substructures based on learned semantic similarity in the latent embedding space, whereas MCS groups molecules purely according to graph-theoretic structural overlap. As a result, VQ could capture higher-level or functional similarities that may not correspond to large MCS fragments.
>
>
>
>
> >**Q1.** *There are two “moderately” words on p.2 line 99 (typo).*
>
> Thank you for pointing this out. We have removed the duplicated word “moderately” in the revised manuscript.
>
> >**Q2.**  *What exact pair embedding is used for the t-SNE plots?*
>
> Thank you for your question. The t-SNE visualizations are generated using the final pair-level embedding produced from the merged graph $\widetilde{\mathcal{G}}$ as defined in Eq. (4). Specifically, after constructing the pairwise merged graph and applying the GNN layers to obtain node embeddings, we aggregate them through the graph-level readout in Eq. (4) to obtain a single vector representation for each molecule pair. These pair embeddings are then using t-SNE for visualization.
>
> >**Q3.** *Which dataset underlies Figure 5a?*
>
> Thank you for the suggestion. The data shown in Figure 5a is derived from the ChchMiner dataset. We will add a clearer explanation in the revised version to clarify the source.
>
> >**Q4.** *Please list the bond/edge features used for intermolecular connections.*
>
> We apologize for the misunderstanding. Our intended meaning is that the merged-graph construction introduces  relational edges that enable the model to learn interaction-relevant structural features. These edges do not represent literal interatomic forces; rather, they capture  dependencies associated with functional groups or atoms that often correlate with hydrogen-bond donor/acceptor behavior or van der Waals compatibility. In the model, these relational edges participate in message passing and update the representations of the connected atoms, thereby enhancing the interaction-aware propagation process. **This is a form of conceptual relational modeling, not physical interaction modeling**.

---

> ### Author Response · Authors · 2025-11-18
> **Response Part V**
>
> >**Q5.** *Model names are missing in Tables 12 and 13—please restore.*
>
> Thank you for pointing this out. Tables 12 and 13 report I2Mole's own performance under varying model capacities and data scalability settings, so the original tables intentionally contained only the results for I2Mole. To avoid confusion, we have now restored the model name explicitly in both tables and adjusted the captions to clearly indicate that these results correspond to I2Mole under different configurations.
>
> >**Q6.** *How exactly are scaffold and size splits implemented?*
>
> Our **scaffold split** follows the standard practice used in molecular OOD evaluation [1,2]. We first match all molecules against a fixed set of SMARTS-defined scaffolds, consisting of nine predefined core substructure patterns as the following Table. All molecules containing any of these scaffolds are assigned to the test set. The remaining molecules are then randomly divided into training and validation sets using a 9:1 ratio. This ensures that structurally distinct scaffolds appear exclusively in the test domain.
>
> For the **size-based** domain shift, following the procedure in [3], we group molecules according to their atomic size (number of atoms). Molecules are sorted in descending order of atomic size, and the ordered sequence is partitioned as follows: the largest 60% are assigned to the training set, the middle 20% to the validation set, and the smallest 20% to the test set.
>
> | `c1ccccc1C(=O)[O&H1]` | `C#[C&H1]`                  | `C[C&H2]Cl`     |
> | `C1=CCCC1`           | `c1ccc2c(-,:c1)ccc1ccccc12` | `C1COCCN1`      |
> | `NS(=O)(=O)C`         | `c1ccccc1[N&+](=O)[O&-]`    | `O[N+]([O-])=O` |
>
>
> [1]  Group contribution and machine learning approaches to predict Abraham solute parameters, solvation free energy, and solvation enthalpy[J].  _JCIM_., 2022, 62(3): 433-446.
>
> [2] MoleOOD: Learning Substructure Invariance for Out-of-Distribution Molecular. _NIPS_. 2024.
>
> [3] DrugOOD: Out-of-Distribution (OOD) Dataset Curator and Benchmark for AI-aided Drug Discovery. _AAAI_., 2023
>
>
> > **Q7.** *For interpretability, specify the matching criteria for “DDI-level, perpetrator-level, frequent functional group,”*
>
> Thank you for the question. In fact, our evaluation protocol follows the matching criteria defined in the MMDDI framework [1]. Specifically：
>
> * **DDI-level matching** evaluates the classification of all 343,036 MMDDIs in the dataset. These consist of 171,518 true interactions, where drug A inhibits or induces the metabolism of drug B, together with an additional 171,518 reverse-order negative samples created by flipping the semantic roles of the two drugs. The model must determine whether the predicted mechanism and direction for each drug pair are correct—that is, whether drug A truly affects drug B.
>
> * **Perpetrator-level matching** evaluates whether the model can correctly determine which drug in the pair is the perpetrator and which is the victim.
>
> * **Frequent Functional Groups Matching** assesses the model’s substructure-level interpretability using a manually curated set of 73 chemicals known from the literature to cause metabolism-mediated DDIs through specific functional groups or substructures. These chemicals collectively form 13,786 relevant DDI pairs.
>
>  We will clarify these criteria explicitly in the revised manuscript to improve readability for the audience.
>
> [1] Learning motif-based graphs for drug–drug interaction prediction via local–global self attention. _Nature machine intelligence_. 2024.
>
> > **Q8.** *How key substructures are selected.*
>
>  Key substructures correspond to the final  subgraph $\widetilde{\mathcal{G}\_{IB}}$. Specifically, key substructures are selected through the noise-injection mechanism in **Section 3.2.2**, each node and relation edge is assigned a learned probability of being replaced by noise. Nodes with higher learned preservation probabilities are less likely to be replaced by injected noise, thus retaining their semantic information throughout optimization. These retained nodes constitute the informative part of the graph and are used to form the final key substructures $\widetilde{\mathcal{G}\_{IB}}$.
>
>
> ---------------
> We greatly appreciate your insightful and helpful comments, as they will undoubtedly help us improve the quality of our article. If our response has successfully addressed your concerns and clarified any ambiguities, we respectfully hope that you consider raising the score. Should you have any further questions or require additional clarification, we would be delighted to engage in further discussion. Once again, we sincerely appreciate your time and effort in reviewing our manuscript. Your feedback has been invaluable in improving our research.

---

> > ### Comment · Reviewer_E6Gc · 2025-11-26
> > **Response to the rebuttal**
> >
> > Thanks for the detailed response. I agree with most points and feel that you have addressed the majority of my concerns. I still have two recommendations:
> >
> > - Narrow the scope to the task most thoroughly studied here—drug–drug interaction prediction—rather than broader “property prediction,” which would be a more rigorous and accurate framing.
> >
> > - While the model captures relational rather than physical interactions, the manuscript currently makes strong physical claims (e.g., hydrogen bonds, hydrophobic effects). Without explicit evidence that the top-n% selected edges correspond to such interactions, these statements are likely to confuse readers, especially given existing work that modeling such interactions based on geometry. I suggest softening or removing those claims.
> >
> > With these clarifications, I will raise my score.

---

> > > ### Author Response · Authors · 2025-11-27
> > >
> > > Thank you very much for your thoughtful suggestions and for your willingness to raise the score. Following your recommendations, we have completed the corresponding revisions in revised version. Specifically, we **narrowed the scope to focus on the DDI task**, which is the core problem studied in this work, and we **removed the physical claims to avoid misunderstanding**. These revisions are primarily reflected in the abstract and the introduction. We sincerely appreciate your constructive feedback, which has helped us improve the rigor and clarity of the manuscript.
> > >
> > > Best,
> > >
> > > The Authors

---

### Official Review · Reviewer_vk1e · 2025-10-31

**Soundness:** 3
**Presentation:** 2
**Contribution:** 2
**Rating:** 4
**Confidence:** 3

**Summary:**

This paper proposes I2Mole, an interaction-aware, invariant molecular relational learning framework for predicting properties of molecular pairs (with a focus on drug–drug interactions).

The key ideas are:
1. build a merged graph that explicitly connects atoms across the two molecules and run intra- and inter-molecular message passing to model atomic interactions
2. extend the Graph Information Bottleneck (GIB) idea to extract an invariant rationale (core subgraph) from the merged graph via controlled noise injection
3. learn a discretized environment codebook (via vector quantization) that represents non-core environmental contexts and is used to encourage invariance / domain robustness.

The authors evaluate on three DDI datasets (ZhangDDI, DeepDDI, ChChMiner) and report consistent gains over baselines in transductive, inductive and domain-shift experiments.

**Strengths:**

* Well-motivated idea. Explicitly modeling inter-molecular atomic interactions addresses a real limitation of existing single-molecule GNNs in pairwise tasks.
* Novel combination of techniques. Extending GIB with a learned VQ codebook for structured noise injection is conceptually interesting and empirically effective.
* Thorough evaluation. The experiments are comprehensive, including multiple datasets, settings, ablations, and statistical significance checks. The fact that performance gains are consistently observed on all settings is especially impressive.

**Weaknesses:**

* Marginal performance improvement. The proposed model is relatively heavy compared with baselines, and some gains (especially on larger datasets) are modest. It’s unclear whether improvements stem mainly from architectural bias or model capacity.
* Presentation quality. The paper is dense and difficult to follow in several parts. Important design choices are buried in the appendix or described briefly. A clearer, higher-level walkthrough and cleaner notation would greatly improve readability. There are also some typos and grammatical errors throughout the text, such as 98-99 "moderately and moderately", 182-183 "$u_i^{\'}$", etc.

**Questions:**

* Could you add some kind of error metrics (such as the standard deviation)  to the reported performance numbers, by training and inferencing the models multiple times with different seeds?
* Could you include parameter-matched baseline comparisons or lighter model variants to better assess the cost–performance tradeoff?
* There are many typos and grammatical errors throughout the text. A careful proofreading could help.
* Please fix the styles for citations (do use `\cite`, `\citet` and `citep` wisely).

---

> ### Author Response · Authors · 2025-11-18
> **Response Part I**
>
> > **W1.** *Marginal performance improvement. It’s unclear whether improvements stem mainly from architectural bias or model capacity.*
>
>
> Thank you for this valuable observation. We would address your concerns regarding performance gain, model complexity, and whether improvements stem from architectural bias or model capacity as follows:
>
> * **On marginal improvements.**  While the absolute gains on large datasets may appear modest, they are consistent across all benchmarks and become particularly pronounced in the **most challenging domain generalization scenarios evaluations which is out key point**. These subsets are well-known to be difficult for existing approaches, and stable improvements in such regimes indicate that our model indeed captures interaction-relevant relational dependencies that baselines fail to model. Importantly, even on large datasets, a gain of ~0.98% is **statistically significant and practically meaningful**, as shown by the significance analysis in Table 8. We will emphasize these results more clearly in the revised version.
>
> * **On model complexity.**  From a model design standpoint, our goal is to explicitly capture the intermolecular relational structure rather than treating the two molecules independently. This represents a key improvement over traditional approaches. As a result, **the added complexity of our model primarily arises from edge-level aggregation and cross-molecule message passing**, since the merged graph naturally contains more relational edges which is an inherent characteristic of GNN-based architectures. Although this makes our model slightly heavier than baselines such as CIGB and CRML, **we conducted additional comparisons with parameter-matched baselines** (e.g., CausalIB, MMGNN, and Explainable GNN). It could be found that our model continues to outperform these baselines (on ZhangDDI dataset) even when they are scaled to the same parameter budget.
>
> | Baseline Model  | Performance (Acc; %) | Parameters (M) |
> | --------------- | --------------- | -------------- |
> | MMGNN           | 85.40           | 389.10         |
> | Explainable GNN | 84.24           | 39.10          |
> | MMHNN           | 86.17           | 32.26          |
> | CasualIB        | 88.14           | 38.17          |
> | I2Mole (Ours)   | 88.64           | 35.40          |
>
> * **Architectural bias vs. model capacity.**  We further conducted ablations to disentangle the roles of architectural bias and pure model capacity. Increasing hidden dimensions or depth alone results in only very minor performance variation (**Appendix Table 12**): within ~5M parameters, the performance fluctuates by <0.5%, and degradation appears only when parameters are aggressively reduced by >10M. In contrast, removing the merged-graph relational edges or the information-bottleneck compression module leads to substantial performance drops (**Table 4**). This clearly shows that the performance improvements predominantly stem from our architectural inductive bias—explicit modeling of intermolecular edges and structured representation compression—rather than from simply adding more parameters.
>
>
>
> [1] Mmgnn: A molecular merged graph neural network for explainable solvation free energy prediction. _IJCAI_. 2024.
>
> [2] Subgraph Information Bottleneck with Causal Dependency for Stable Molecular Relational Learning. _IJCAI_. 2025.
>
> [3] Molecular merged hypergraph neural network for explainable solvation gibbs free energy prediction.  _Research_. 2025.
>
> [4] Explainable Solvation Free Energy Prediction Combining Graph Neural Networks with Chemical Intuition. _JCIM_. 2022.
>
> >**W2.** *Presentation quality. A clearer, higher-level walkthrough and cleaner notation would greatly improve readability.*
>
>
> Thank you for the helpful feedback. We have conducted extensive experimentation and analysis for this work. Due to ICLR’s 9-page limit, many important components—including the detailed ablations, theoretical proofs, full hyperparameter settings, and additional visualizations—had to be placed in the appendix. These sections are intended as essential extensions of the main text. To improve readability, we have made several revisions in revised version:
>
> * Added a structured appendix table of contents to make navigation of key components (design choices, proofs, hyperparameters, ablations, and visualizations) much clearer.
> * Improved the high-level explanation of the core architecture in the main text to reduce conceptual density.
> * Refined notation and reorganized some descriptions to make the main pipeline easier to follow.
>
> If there are specific parts that felt unclear or hard to follow, we would be grateful for pointers—we are fully committed to improving the clarity of the manuscript.

---

> ### Author Response · Authors · 2025-11-18
> **Response Part II**
>
> >**W2&Q3.** *There are also some typos and grammatical errors throughout the text, such as 98-99 "moderately and moderately",182-183", etc.*
>
> Thank you for pointing this out. We have corrected the typos and grammatical errors mentioned (e.g., lines 98–99 and 182–183), and we have conducted a thorough proofreading of the entire manuscript to ensure overall linguistic clarity and consistency.
>
>
> > **Q1.**  *Could you add some kind of error metrics (such as the standard deviation) to the reported performance numbers, by training and inferencing the models multiple times with different seeds?*
>
>
> Thank you for the valuable suggestion. We apologize for not making this clearer in the main text. As noted in Line 373, all reported results are already averaged over eight independent runs with different random seeds. The numbers in parentheses in the bottom-right corner of each table cell represent the standard deviation across these runs. To avoid any ambiguity, we will explicitly clarify this in the table captions in the revised manuscript version.
>
>
> >**Q2.** *Could you include parameter-matched baseline comparisons or lighter model variants to better assess the cost–performance tradeoff?*
>
>
> Thank you for the helpful suggestion. In the original submission, we included comparisons with baselines of varying capacities (Table 14). Following your advice, **we additionally conducted parameter-matched comparisons**, where we scale models such as CausalIB, MMGNN, and Explainable GNN to comparable parameter budgets. Across all configurations, I2Mole consistently outperforms these parameter-matched baselines, indicating that the observed improvements cannot be attributed simply to increased model size.
>
> | ZhangDDI dataset |                 |                |
> | ---------------- | --------------- | -------------- |
> | Baseline Model   | Performance (Acc; %) | Parameters (M) |
> | MMGNN            | 85.40           | 389.10         |
> | MMHNN            | 86.17           | 32.26          |
> | CasualIB         | 88.14           | 38.17          |
> | I2Mole (Ours)    | 88.64           | 35.40          |
>
> The higher computational cost of I2Mole primarily stems from edge-level aggregation and cross-molecule message passing, since the merged graph naturally contains more relational edges—an inherent property of graph neural networks that explicitly model relational structure, rather than a consequence of unnecessary parameter expansion. Furthermore, **we have also explored several lightweight variants** in the original manuscript (Table 17). Specifically, we replaced the intermolecular message-passing module with different lightweight GNN architectures. As summarized in the paper, these replacements significantly reduce the number of parameters and computation, but also lead to notable drops in performance. This further confirms that the relational message-passing design is crucial for capturing intermolecular dependencies.
>
> | Model Variant      | Performance (Acc; %) | Parameters (M) |
> | ------------------ | --------------- | -------------- |
> | MPNN->>GraphSAGE   | 80.26           | 21.30          |
> | MPNN->>GAT         | 87.42           | 22.40          |
> | MPNN->>1-layer GIN | 95.07           | 31.40          |
> | MPNN->>2-layer GIN | 94.98           | 26.50          |
> | MPNN->>3-layer GIN | 91.35           | 21.40          |
> | MPNN->>HyperGraph  | 92.37           | 13.225         |
> | I2Mole (Ours)      | 95.34           | 35.40          |
>
> We will incorporate these additional results and clarifications into the revised manuscript.
>
> [1] Mmgnn: A molecular merged graph neural network for explainable solvation free energy prediction. _IJCAI_. 2024.
>
> [2] Subgraph Information Bottleneck with Causal Dependency for Stable Molecular Relational Learning. _IJCAI_. 2025.
>
> [3] Molecular merged hypergraph neural network for explainable solvation gibbs free energy prediction.  _Research_. 2025.
>
> >**Q4.** *Please fix the styles for citations (do use `\cite`, `\citet` and `citep` wisely).*
>
> Thank you for the valuable suggestion. In the current submission, we used `\cite` uniformly for all references. We appreciate your recommendation that using `\cite`, `\citet` and `citep`appropriately can improve readability and better follow standard LaTeX citation practices. This is very helpful and professional, and we have updated the citation styles in the revised version accordingly.
>
> ---------------
> We greatly appreciate your insightful and helpful comments, as they will undoubtedly help us improve the quality of our article. If our response has successfully addressed your concerns and clarified any ambiguities, we respectfully hope that you consider raising the score. Should you have any further questions or require additional clarification, we would be delighted to engage in further discussion.

---

> ### Comment · Reviewer_vk1e · 2025-11-25
>
> I thank the authors for their prompt response.
>
> ---
>
> **W1**: Thank you for the clarifications. To my eyes, however, the improvements are still not strong enough. They might be statistically significant, but not substantial. In my opinion, the complexity of the method should match with the performance improvement brought by it. That is, the most impactful works are those with simple, transferable techniques and strong results (e.g., DropOut, ResNet, etc.). The second tier works are either ones with simple techniques and less strong results, or ones with complicated implementations and strong results. The present work, I2Mole, has a complicated implementation but marginal performance improvements. This is usually not preferable.
>
> ---
>
> **W2**: Am I missing something? I cannot find the new revision of the manuscript. Did you forget uploading?
>
> ---
>
> **Q2**: Is there something wrong with the "MPNN->>GIN" results? The models with more layers of GIN have fewer parameters, which is not sensical.
>
> ---
>
> By the way, your response should be made readable to "Everyone".

---

> > ### Author Response · Authors · 2025-11-25
> > **Response to Reviewer vk1e**
> >
> > Thank you very much for your prompt reply.  Regarding the remaining issues you mentioned, we would like to provide further clarification.
> >
> > > **W1.** *The improvements is statistically significant, but not substantial.*
> >
> > We sincerely thank the reviewer for the constructive comments, and we appreciate the acknowledgment that I2Mole achieves a significant improvement over the baselines. We would like to emphasize that the central motivation of I2Mole is OOD generalization, not merely improving in-distribution accuracy. Therefore, the proper evaluation should focus on the OOD test results, where our model shows clear and substantial gains.
> >
> >  > **W1.** *I2Mole has a complicated implementation.*
> >
> >  In terms of the parameters problem, I sincerely agree your perspective that simple and transferable techniques are often the most elegant contributions. To be honest, I am appreciative of your high academic taste and discernment. However, I would like to point out that:
> >
> >
> > * Even widely used architectures such as ResNet-50 contain around 25M parameters, and commonly adopted Transformer-based models typically range from 80M to 100M parameters. In comparison, the 35M parameters of I2Mole are modest by today’s standards. If cutting-edge performance could be achieved with a one-line code, it would indeed be exciting, but such breakthroughs are extremely rare in the decades-long evolution of AI.
> >
> >
> > * Within the field of DDI prediction, our initial choice of baselines already leaned toward relatively small models. Even under this setting, I2Mole does not exceed these baselines by an order of magnitude (Table 14). After incorporating the more recent models mentioned in the revised version, we further confirmed that the parameter size of I2Mole is actually smaller than the latest models such as CausalIB and MMGNN. Objectively speaking, a model with 35M parameters is not considered complex in modern AI practice.
> >
> > * The emergence of AI methods has continuously advanced their respective fields in various ways. Though model size is certainly an aspect worth considering, scientific progress relies far more on accumulated exploration and gradual refinement. Scientific breakthroughs do not appear overnight, ResNet, Dropout, and even Transformers are all the result of years of deep, sustained effort by their creators groups. Although these models now shine brightly, many lesser-known works that quietly pushed the field forward have also played crucial roles and deserve recognition and fair evaluation.
> >
> > By the way, speaking a bit philosophically, your viewpoint reflects an idealist interpretation of scientific progress, and I truly appreciate this way of thinking. I look forward to continue this discussion further.
> >
> >
> >
> >  > **W2.** *New revision of the manuscript.*
> >
> >
> > We are currently revising the main text, and we hope to finalize the updated version once we are confident that our responses have fully addressed the reviewer’s concerns. This approach helps avoid unnecessary back-and-forth revisions and minimizes the number of times you need to re-review the manuscript. Given your clear emphasis on efficiency and  high academic taste, we trust this intention resonates with you. We believe we have clearly communicated all changes and ensured that they comply with ICLR’s policy requirements.  If you have further questions, please let me know promptly, and we will complete the revision process together. Nevertheless, we will promptly upload the revised version as soon as you are satisfied with the clarifications provided.
> >
> >
> >  > **Q2.** *Is there something wrong with the "MPNN->>GIN" results? The models with more layers of GIN have fewer parameters, which is not sensical.*
> >
> >
> >  Thank you for this interesting question. The answer can be found in the main manuscript at Line 1161. Our architecture uses a 3-layer MPNN, and in the experiments we replaced these layers with different numbers of GIN layers.
> >
> >
> >  >*Your response should be made readable to "Everyone".*
> >
> >  OK，I will do it latter.
> >
> > ---
> > We greatly appreciate your insightful and helpful comments, as they will undoubtedly help us improve the quality of our article. If our response has successfully addressed your concerns and clarified any ambiguities, we respectfully hope that you consider raising the score. Should you have any further questions or require additional clarification, we would be delighted to engage in further discussion.

---

> > > ### Comment · Reviewer_vk1e · 2025-11-28
> > >
> > > I thank the authors for the response.
> > >
> > > ---
> > >
> > > **W1**: Sorry, I did not make my point clear. In my initial review, what concerns me was the model capacity, which can be quantified by the number of parameters. In the response, what I meant was the **difficulty of implementing the proposed techniques and/or further improving upon it**.
> > >
> > > However, the results in your response to Reviewer E6Gc (https://openreview.net/forum?id=IqwF00TCmf&noteId=esmvarwZBa) show substantial improvement of I2Mole over the baseline models, albeit details of the datasets/tasks are not provided. These results have increased my belief that I2Mole can lead to substantial improvements on some tasks. Why are those results not incorporated into the manuscript?
> > >
> > > ---
> > >
> > > **Q2**: I see. I was misled by the "->>" notation in the table in the first response. This issue does not occur in the manuscript, so never mind.
> > >
> > > ---
> > >
> > > **On the newly revised manuscript**:
> > >
> > > 1. Putting acronyms (e.g., DDI) in the title is not encouraged.
> > > 2. I can see that most presentation issues have been improved.

---

> > > > ### Author Response · Authors · 2025-11-28
> > > >
> > > > >**W1：** *Why are those results not incorporated into the manuscript?*
> > > >
> > > > Thank you for the suggestion. We originally added those experiments to demonstrate that I2Mole can work on other molecular interaction scenarios, but in response to reviewers E6Gc’ recommendations, we narrowed the scope of the paper to DDI prediction, which is the core task of this work. Therefore, those additional experiments were not included in the main paper at that time.
> > > >
> > > > Following your advice, we have now added these results to the Appendix (Table 15) in the latest version, along with the corresponding explanations to ensure completeness and transparency.
> > > >
> > > > >**Newly revised manuscript-1：** *Putting acronyms (e.g., DDI) in the title is not encouraged.*
> > > >
> > > > Thank you for the professional suggestion. We have revised the title accordingly.
> > > >
> > > > >**Newly revised manuscript-2：** *I can see that most presentation issues have been improved.*
> > > >
> > > > We sincerely thank the reviewer for the continued engagement and the constructive follow-up comments. We appreciate the clarification regarding your original concerns, and we are grateful that the additional results strengthened your confidence in the potential impact of I2Mole. We deeply appreciate the time and effort you have devoted to this review process.
> > > >
> > > > -----
> > > > Given that we have carefully addressed all concerns raised in your earlier reviews, we respectfully hope that you consider a positive score. Should you have any further questions or require additional clarification, we would be delighted to engage in further discussion. Once again, we sincerely appreciate your time and effort in reviewing our manuscript. Your feedback has been invaluable in improving our research.

---

### Author Response · Authors · 2025-12-01
**Author Final Remarks by Authors**

Dear (senior) AC,

We extend our sincere gratitude for the time and effort you have dedicated to review our manuscript. We greatly enjoy the in-depth discussions with the reviewers and appreciate their valuable feedback. Here, to facilitate your efficient assessment, we have summarized the essential revisions and responses into a concise table for your quick review, aiming to save your time and effort.

| **Reviewer** | **Main Concerns**                                                                                                                             | **Our Response**                                                                                                                                                                                               | **Outcome**                            |
| ------------ | --------------------------------------------------------------------------------------------------------------------------------------------- | -------------------------------------------------------------------------------------------------------------------------------------------------------------------------------------------------------------- | -------------------------------------- |
| `vk1e`       | typos, performance improvement; more error metrics; parameter-matched baseline comparisons; lighter model variants. **(Round I)**             | added statistically significant analysis; added the latest models to compare; added several lightweight variants;  Conducted a full word-by-word revision;                                                     | Further discussion                     |
|              | needs upload updated revision,  concern about the complicated implementations, parameters problems. **(Round II)**                            | clarified the model's contribution; explained the parameter issue; uploaded a new version                                                                                                                      | Further discussion                     |
|              | acronyms need to revise; added experiments should incorporate into the manuscript **(Round III)**                                              | revised the manuscript and upload                                                                                                                                                                              | **Positive feedback.**                 |
| `E6Gc`       | scope misalignment; overstated physical claims; result consistency issues; missing experimental details;  typographical Issues<br>            | added extra experiments; clarified the role of relationship edges; revised the manuscript; completed scope clarification; enhanced interpretability analysis with MCES result; present experimental details    | **Score improved, Positive feedback.** |
| `WU3y`       | scientific contribution further clarity; typographical Issues; justify physical claims; explain the concept of molecular environment codebook | clarify the scientific contribution; correct the formatting and grammatical issues; explain the concept of environment codebook; clarified the role of relationship edges;                                     | **No feedback until now. (boardline score)**             |
| `qi4Q`       | Missing standardized DDI benchmarks; Expanding evaluation scope; insufficient physicochemical mechanism analysis; typo issues                 | Added _DrugBank_ and _TWOSIDES_ benchmarks; compared hit rates of metabolism-mediated DDI substructures for interpretability; analysis the physicochemical mechanisms with the models; revised the manuscript. | **Positive feedback. Tends to Accept.**         |

------
Thank you very much for your patience and the considerable effort you have devoted to our work. **We truly value this opportunity to present our work at ICLR and sincerely hope to earn the AC’s understanding and support.** Regarding the recent accident, we would like to clarify that we have never engaged in any form of collusion with reviewers nor had any such intention. **We fully understand the seriousness of the matter. We respectfully assure you of the integrity and independence of our submission**.

Best regards,

Authors

---

### Meta-Review · Area_Chair_LLQF · 2026-01-06

**Summary:**

This paper proposes I2Mole, an interaction-aware, invariant molecular relational learning framework for predicting drug-drug interactions by constructing a merged molecular graph, applying a Graph Information Bottleneck to extract invariant core subgraphs, and learning a discretized environment codebook to encourage domain robustness.

**Reviewer Concerns:**

The idea is considered well-motivated and addresses a key limitation of single-molecule GNNs for pairwise tasks (vk1e), a novel and clearly specified combination of techniques (E6Gc), and methodologically well-structured with comprehensive experiments (qi4Q). Although, some weaknesses remain, including modest performance gains relative to model complexity, overstatement of claims regarding general molecular property prediction and physical interaction modeling, presentation issues, and a lack of evaluation, fortunately, the authors have addressed the main issues in their response.

**Reviewer Scores:**

Reviewer vk1e and E6Gc, having their key concerns addressed, would likely raise their scores to acceptance.

Reviewer WU3y would remain the score.

Reviewer qi4Q, satisfied with the additional experiments, would lean toward acceptance.

---

### Decision · Program_Chairs · 2026-01-26

Accept (Poster)